# Higher energy and safer sodium ion batteries via an electrochemically made disordered $Na_3V_2(PO_4)_2F_3$ material

Guochun Yan[1,2,7], Sathiya Mariyappan[1,2], Gwenaelle Rousse[1,2,3], Quentin Jacquet[1], Michael Deschamps[2,4], Renald David[5], Boris Mirvaux[1,2], John William Freeland[6] & Jean-Marie Tarascon[1,2,3]

The growing need to store an increasing amount of renewable energy in a sustainable way has rekindled interest for sodium-ion battery technology, owing to the natural abundance of sodium. Presently, sodium-ion batteries based on $Na_3V_2(PO_4)_2F_3$/C are the subject of intense research focused on improving the energy density by harnessing the third sodium, which has so far been reported to be electrochemically inaccessible. Here, we are able to trigger the activity of the third sodium electrochemically via the formation of a disordered $Na_xV_2(PO_4)_2F_3$ phase of tetragonal symmetry ($I4/mmm$ space group). This phase can reversibly uptake 3 sodium ions per formula unit over the 1 to 4.8 V voltage range, with the last one being re-inserted at 1.6 V vs $Na^+/Na^0$. We track the sodium-driven structural/ charge compensation mechanism associated to the new phase and find that it remains disordered on cycling while its average vanadium oxidation state varies from 3 to 4.5. Full sodium-ion cells based on this phase as positive electrode and carbon as negative electrode show a 10–20% increase in the overall energy density.

[1] Chimie du Solide-Energie, UMR 8260, Collège de France, 75231 Paris Cedex 05, France. [2] Réseau sur le Stockage Electrochimique de l'Energie (RS2E), Cedex FR CNRS 3459, Amiens 80039, France. [3] Sorbonne Université - 4 Place Jussieu, 75005 Paris, France. [4] CNRS, CEMHTI UPR3079, Univ. Orléans, Orléans 45100, France. [5] LRCS, Université de Picardie Jules Verne, 80039 Amiens, France. [6] Advanced Photon Source, Argonne National Laboratory, Argonne 60439 IL, USA. [7] Present address: School of Metallurgy and Environment, Central South University, Changsha 410083, China. Correspondence and requests for materials should be addressed to J.-M.T. (email: jean-marie.tarascon@college-de-france.fr)

The development of new types of high-performance energy storage and conversion technologies is urgently needed to meet the growing demands for portable electronic equipment, electric vehicles, and large-scale smart grids[1,2]. Batteries, as one of the most versatile energy storage technologies, play a central role in the transition from fossil-based fuels to renewable energy. While the Li-ion technology is a key enabler in the transport sector, it falls short in the stationary storage sector because of high cost linked to the limited abundance of lithium. Sodium-ion battery technology has recently aroused great interest, among all the scientific community, as a valid and more environmentally friendly alternative to Li-ion, owing to the abundance of sodium all over the planet[3,4]. Present sodium ion systems rely on carbon as the negative electrode and of either Na-based layered oxides or polyanionic compounds as the positive electrode[5–9]. Through comparative studies, it has been demonstrated that the $Na_3V_2(PO_4)_3$/C system presently offers both cycling stability and power rate advantages over systems using the Na-based layered oxide $Na_xMO_2$ (where $x \leq 1$ and M = transition metal ion(s))[9–12].

Thus, this paper further focusses on the polyanionic $Na_3V_2(PO_4)_2F_3$ positive electrode material, now on termed as NVPF, from which one can reversibly remove two sodium ions per formula unit via two-step redox plateaus of equal amplitudes centered at ~3.7 and ~4.2 V vs. $Na^+/Na^0$. Besides, NVPF electrodes offer a sustained reversible capacity of 128 mAh g$^{-1}$ together with a specific energy of 500 Wh kg$^{-1}$, while showing excellent capacity retention and rate capability[9,13,14]. However, to make sodium ion cells based on NVPF technologically relevant, there is a need to increase their specific energy, which is less competitive than today's Li-electrodes (~600 Wh kg$^{-1}$ for LiCoO$_2$-type materials), hence the various attempts to boost the capacity of NVPF electrodes. A first approach has consisted in the successful and reversible electrochemical insertion of 0.5 sodium ion in $Na_3V_2(PO_4)_3$ or 1 sodium ion in $Na_3V_2(PO_4)_2O_2F$ at potentials of ~1.6 V, respectively[15–17]. Such a low potential limits the specific energy gain associated to the insertion of extra sodium ions but in contrast provides the feasibility to use $Na_{3+x}V_2(PO_4)_2F_3$ composites as a sodium reservoir, as previously demonstrated, to compensate for the sodium loss at the carbon negative electrode during the first cycling[15].

Another obvious path to increase the specific energy of NVPF consists in harnessing the remaining sodium ($Na_1V_2(PO_4)_2F_3$–$Na_0V_2(PO_4)_2F_3$) at high potential so as to reach theoretical energy densities of ~800 Wh kg$^{-1}$. A significant amount of research effort has been directed toward this goal, but have remained unproductive in accordance with density functional theory calculations stating that the removal of the third sodium ion should occur at potential (>4.9 V) that is too high for present electrolytes[18,19]. The accessibility of the third sodium ion in NVPF was thus remaining an open question.

Inadvertently, in our search toward exploring better electrolytes for the NVPF/C sodium ion system[20] we observed, by prolonged charging time at high potential, the feasibility to modify the voltage-composition profile of the charge/discharge curve (e.g smoothing of the voltage features), hence providing a hint of some sodium electrochemical activity at high potential (Supplementary Figure 1). This inspired us to undertake a deeper exploration of $Na_3V_2(PO_4)_2F_3$ oxidation at high voltage.

Here in this work, we demonstrate the feasibility to electrochemically remove nearly three sodium ions upon oxidation till 4.8 V vs $Na^+/Na^0$ with the concomitant formation of a new disordered "NVPF" phase that can reversibly uptake and release around three sodium ions on the following cycle; two between 4.2 and 3.6 V and the last one at 1.6 V. This provides a 20% gain in specific energy for NVPF/C sodium ion cells. We also demonstrate the benefits of the low voltage plateau to secure the use and the storage of such cells down to zero volts, and the feasibility of monitoring the state of charge (SoC) thanks to the S-shape profile of the voltage-composition curves. Our findings offer unprecedented insight into the development of highly performing sodium ion systems.

## Results

**Activating the third sodium in $Na_3V_2(PO_4)_2F_3$ electrochemically.** Pristine single-phase $Na_3V_2(PO_4)_2F_3$ material, as defined by complementary X-ray diffraction (XRD), scanning electron microscopy, and inductive plasma analysis (ICP), was prepared via a two-step procedure as described in ref. [21]. Figure 1 compares the voltage-composition curves and cycling performances of various NVPF/Na cells charged by limiting the amount of extracted sodium ions ($\Delta x$) to 2.0, 2.25, 2.5, 2.75, and 3.0 to produce samples that from now on will be referred to as NVPF-2, NVPF-2.25, NVPF-2.5, NVPF-2.75, and NVPF-3.0 respectively, and then discharged to 3.0 V. The profile difference between the first charge and subsequent discharge curve increases with $\Delta x$ and becomes the most pronounced for $\Delta x = 3.0$. This suggests an electrochemical-driven irreversible structural change during the first charging process, but once the first cycle is achieved, subsequent charges and discharges superimpose. Note also a smoothing of the stair-case discharge curve for $\Delta x = 2.0$ (Fig. 1a left) with increasing $\Delta x$ that can be equally visualized on the corresponding derivative d$Q$/d$V$ plots (Fig. 1a right). Such a shift from a stair-case to an S-type voltage profile is usually viewed as a positive asset for better SoC monitoring of the cell via the battery management system while obviously the decrease in potential is viewed as a negative asset for high-energy density. Whatever the amount of extracted sodium, only two sodium ions can be reversibly inserted, or slightly less for the $\Delta x = 3.0$ sample that also shows the more pronounced capacity decay (Fig. 1b). Note that in neither case the kinetics of the reversible process seems to be affected as the power rate traces neatly superimpose irrespective of $\Delta x$ (see Supplementary Figure 2). The sodium content for the various $\Delta x$ samples at the end of charge was determined both by Electron Diffraction X-ray analysis (EDX) (Supplementary Figure 3) and ICP on electrodes recovered from the cells, which were washed and dried. These values converge with those deduced from coulometric titration at ±7%, overall three sodium ions are extracted on charge while solely two can be reinserted down to 3.0 V, hence the question of the missing one sodium ion.

To address this point, another set of five NVPF/Na cells were charged identically as the first ones but discharged to 1.0 V. Importantly, the amplitude of the low voltage plateau (1.6–1.3 V) previously spotted for NVPF (here NVPF-2) is now enlarged with increasing $\Delta x$ (Fig. 2a). Strikingly, its increasing amplitude nicely matches with the extra amount of sodium ions extracted beyond $\Delta x = 2.0$ as shown in the inset of Fig. 2a, as if there was a transfer from the high (~4.75 V) to the low voltage plateau (~1.6 V). When cycled between 4.4 and 1.0 V, the NVPF-3.0 sample shows a reversible electrode capacity of 200 mAh g$^{-1}$ (Fig. 2b) as compared to 107 mAh g$^{-1}$ when the cycling is limited between 4.4 and 3.0 V, in addition a more sustained capacity retention for the larger (4.4–1.0 V) cycling voltage range (Fig. 2c vs. Fig. 1b). This ~40 % gain in capacity translates into solely ~15% benefit in energy density because most of the extra discharge capacity is delivered at a low potential (Supplementary Figure 4). Still, this new polymorph exhibits the highest energy density reported so far among the NVPF compounds or their oxygenated variants (see Supplementary Table 1).

At this stage, a delicate question is to understand the origin of the large voltage drop when inserting the third sodium ion on

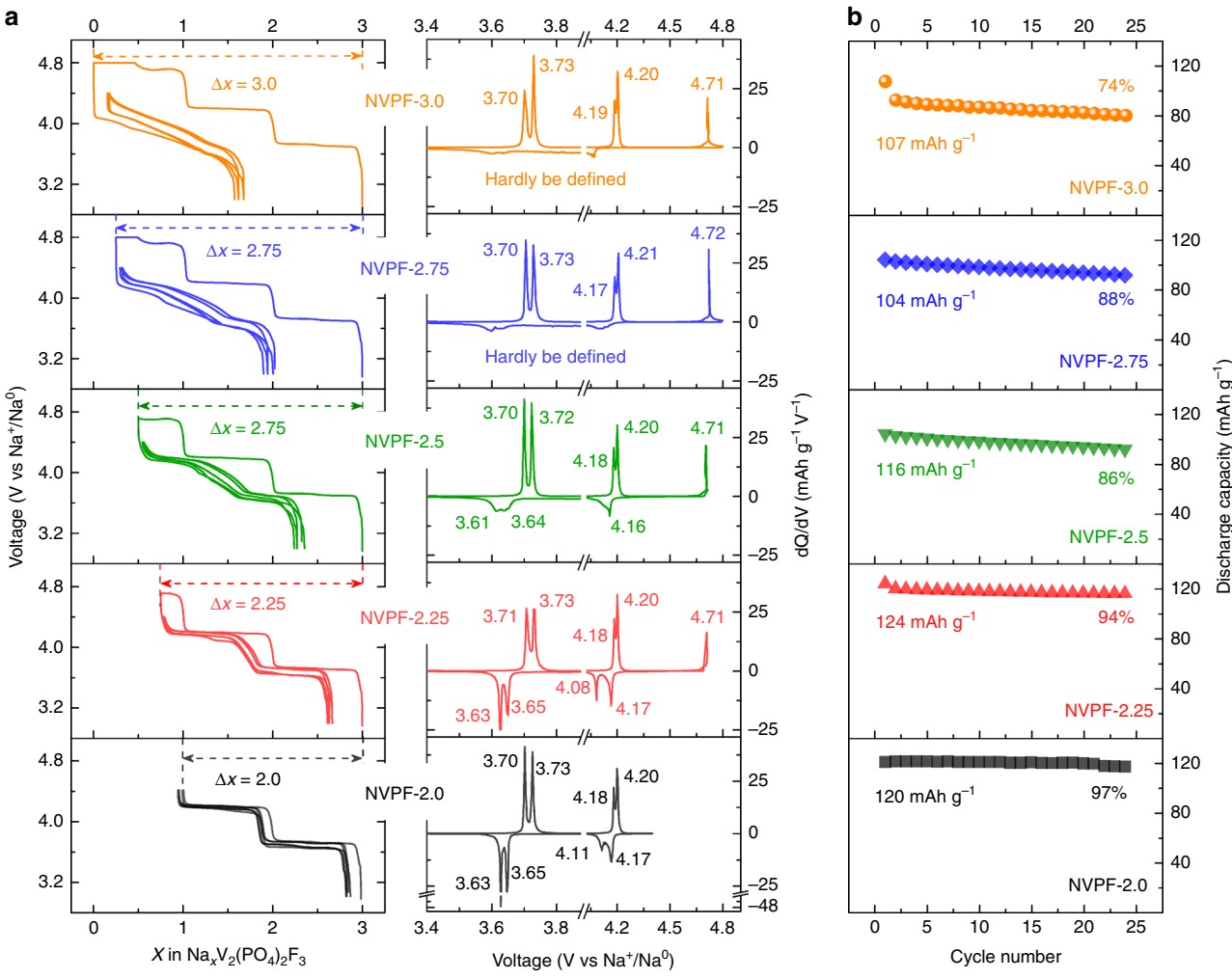

**Fig. 1** Electrochemical characterization of NVPF samples in NVPF/Na half cells at a current of C/10 and cycled between 4.4 and 3.0 V. The first charge process alone is controlled by limiting the amount of Na$^+$ extracted ($\Delta x = 2.0$, 2.25, 2.50, 2.75, and 3.0). **a** Voltage-composition curves (left) and their corresponding d$Q$/d$V$ curves (right). **b** The corresponding capacity retention plots

discharge. Obviously it is not related to the vanadium redox couple as it occurs upon reduction of both Na$_3$V$_2$(PO$_4$)$_2$F$_3$ and Na$_3$V$_2$(PO$_4$)$_2$FO$_2$ independently of the involved couples are V$^{3+}$/V$^{2+}$ and V$^{4+}$/V$^{3+}$[15,16]. In contrast, previous studies have proposed that it could be due to changes in sodium mobility/diffusivity[19,22]. To check this point, galvanostatic intermittent titration-technique (GITT)-type measurements, which combine current pulses and open-circuit sequences, were performed during the second cycle for the NVPF-2.75 (Fig. 3). From the potential jump observed immediately after the application of the current pulse by ~1 s, we could deduce the variation of the cell resistance, which encompasses short-time-constant phenomena, such as electrolyte, electronic contact resistance, and charge transfer resistances. It shows an increase during discharge that is not correlated with the sudden potential drop. Moreover, changing the carbon concentration (10 or 50%) had no effect on the potential drop, therefore definitely ruling out possible electronic limitations across the conductive matrix of the electrode, as shown in Supplementary Figure 5. In contrast, there is just after the potential drop (Fig. 3), a markedly increase of the potential relaxation during the open circuit voltage (OCV) steps, which indicates a long-time-constant phenomena corresponding to the

slowed down diffusion of sodium ion in the solid phase. Since kinetics does not explain the voltage drop, it likely originates from thermodynamics. The plot of the potential after relaxation as a function of sodium stoichiometry (shown as a dashed line in Fig. 3) shows a hysteresis with different traces on charge and discharge, which point toward different reaction pathways. Thus, the question arises on how the Na$_x$V$_2$(PO$_4$)$_2$F$_3$ structure accommodates the uptake and removal of sodium ions.

**Structural evolutions when activating the third sodium in Na$_3$V$_2$(PO$_4$)$_2$F$_3$.** To address this point, ex situ synchrotron XRD measurements were performed on samples recovered from cells that were charged to $\Delta x = 2.0$, 2.25, 2.5, 2.75, and 3.0 and on two other sets of cells that were similarly charged and subsequently discharged to 3 and 1 V, respectively. The powder was placed in 0.7 mm glass capillaries and XRD patterns, as shown in Fig. 4, were recorded in transmission mode at the 11BM synchrotron beamline at Argonne National Laboratory with a wavelength of 0.412 Å. The pristine Na$_3$V$_2$(PO$_4$)$_2$F$_3$ sample can be refined successfully with the Rietveld method (Supplementary Figure 6) in the orthorhombic *Amam* space group with lattice parameters

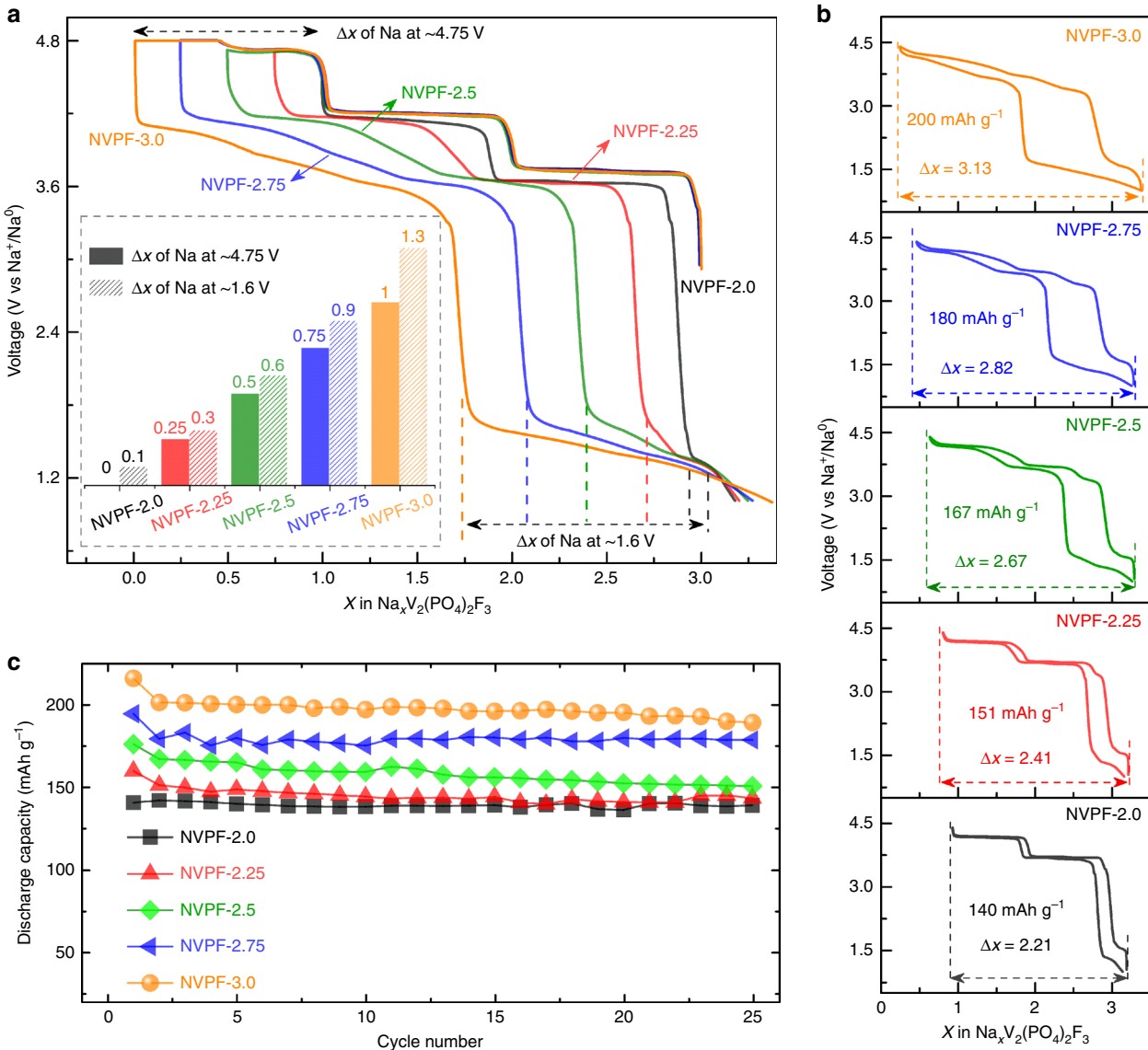

**Fig. 2** Electrochemical characterization of NVPF samples in NVPF/Na half cells cycled between 4.4 and 1.0 V. **a** First cycle activation in which the charging potential is used up to 4.8 V and charge process controlled by limiting the $\Delta x$ (Na) = 2.0, 2.25, 2.50, 2.75, and 3.0 extracted, followed by discharge to 1 V at C/10. The inset shows the amount of $Na^+$ extracted at ~4.75 V and reinserted at ~1.6 V. **b** The charge-discharge profile of the subsequent second cycle in the voltage window of 4.4 and 1.0 V. **c** The corresponding capacity retention plots

$a = 9.02976(14)$ Å, $b = 9.04367(14)$ Å, and $c = 10.75371(9)$ Å, in agreement with previous reports[21,23]. Partial views of the refined structure are shown in Fig. 5. NVPF exhibits a three-dimensional structure built on $V_2O_8F_3$ bi-octahedra bridged by $PO_4$ tetrahedra, and sodium occupies three interstitial sites: Na1 (pyramidal) fully occupied, and Na2 (pyramidal) and Na3 (capped prism) being partially occupied. In this structure, V is distributed on two crystallographic sites, which are similar in terms of environment and distances, and consistent with $V^{3+}$ (average distance 1.98 Å, Fig. 5).

The charged samples show a pronounced evolution of XRD patterns with increasing $\Delta x$ that can easily be spotted at low $\theta$ angles (Fig. 4a left). All Rietveld refinements are shown in Supplementary Figure 7 with the corresponding Supplementary Tables 2 to 7 gathering all deduced structural parameters. First, let us note that the observed phase upon sodium extraction until $\Delta x = 2$ perfectly agrees with early literature report[23]. $Na_1V_2(PO_4)_2F_3$

presents a pattern that can be indexed in a $Cmc2_1$ space group with lattice parameters $a = 8.81577(19)$ Å, $b = 8.8288(3)$ Å, and $c = 11.00215(16)$ Å ($V = 856.32$ Å$^3$ and $V/Z = 214.08$ Å$^3$). In this structure, vanadium atoms are distributed on two crystallographic sites, each corresponding to $V^{3+}$ and $V^{5+}$ oxidation states, while sodium occupies a single pyramidal site with full occupancy (Fig. 5). The further oxidized sample ($\Delta x = 2.25$) shows a similar pattern that also can be indexed in $Cmc2_1$ with however slight changes in lattice parameters ($a = 8.8416(4)$ Å, $b = 8.8567(5)$ Å, and $c = 10.9804(3)$ Å), which reflects a decrease of sodium occupancy. For $\Delta x = 2.5$ and above, there is appearance of new peaks that progressively grow when $\Delta x$ is increased, reflecting the presence of a mixture of the $Cmc2_1$ phase and a new one. For $\Delta x = 3$, this new phase appears as a single phase with an approximate chemical composition $Na_0V_2(PO_4)_2F_3$ and its XRD pattern can be indexed with a tetragonal cell in space group $I4/mmm$, and lattice parameters $a = 6.19887(18)$ Å and $c = 11.3865(8)$ Å. The unit cell

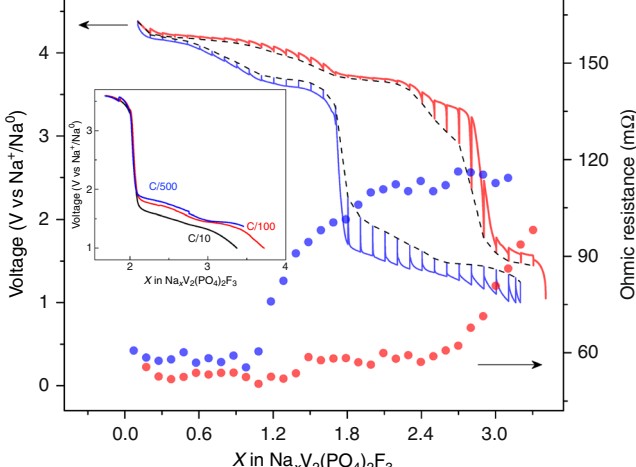

**Fig. 3** Galvanostatic intermittent titration-technique (GITT) test of NVPF-2.75 sample after the first formation cycle. The cell was cycled at C/10 rate and the relaxation time was controlled either by 4 h or by limiting d$V$/d$t$ < 0.1 mV/s; The red and blue solid lines show the experimental GITT curve on charge and discharge respectively, with the black dashed line showing the equilibrium potential after each relaxation process. The red and blue filled circles represent the Ohmic resistance at each point on charge and discharge process respectively. The inset shows NVPF-2.75/Na cell cycled at C/10, C/100, and C/500 within the range of voltage drop during the first discharge process

volume is 437.53(3) Å$^3$ and corresponds to $V/Z$ = 218.76 Å$^3$. This increase, compared to the sample for $\Delta x = 2$, arises from a larger $c$ lattice parameter reminiscent of stronger electrostatic repulsions that are no longer screened by sodium atoms. The Rietveld refinement for $\Delta x = 3$ is shown in Supplementary Figure 7 with the resulting structural parameters listed in Supplementary Table 3. This structural description enlists only one crystallographic site for V, and refined V–O and V–F bond lengths (average bond length 1.87 Å) indicate an oxidation state of V in agreement with the expected value of 4.5+, as also confirmed by bond valence sum calculations. Through discharge down to 1 V, the $\Delta x$ = 2.0, 2.25, and 2.5 phases convert back to the *Amam* structure of the pristine material further confirming the reversibility of the system within the 4.4–1.0 V voltage range (Fig. 4b). Interestingly, the $\Delta x$ = 2.75 and 3 samples stand as exceptions, since they never convert back into the pristine *Amam* phase as shown in Fig. 4b and Supplementary Figure 8. On the opposite, their patterns can be indexed in the tetragonal *I4/mmm* space group, as for the charged phases, and having sodium in a disordered state (i.e. Na occupies several Wyckoff sites with partial occupancy). These disordered reduced phases present the same symmetry and crystal structures (Na distribution) as the high temperature form of $Na_3V_2(PO_4)_2F_3$[21]. In this structure, Na atoms occupy two crystallographic sites (Wyckoff sites 8$h$ and 16$l$) that form a circle around fluorine atoms (Fig. 5). Refining occupancies lead to 2.4 Na per formula unit when the discharge is limited to 3 V, and 3.25 Na when the discharge is pursued down to 1 V (see Supplementary Tables 4 and 5). Consistently, the unit cell volume of the disordered phase $V/Z$ = 222.87 Å$^3$ is larger than that of the pristine one $V/Z$ = 219.54 Å$^3$ and of NVPF-2.0 discharged back to 1 V ($V/Z$ = 220.57 Å$^3$), respectively.

**Evidence of sodium disorder in the newly formed tetragonal phase**. To further characterize the extent of disorder in the structure and its origin, we recorded the $^{23}$Na and $^{31}$P magic

angle spinning-nuclear magnetic resonance spectra of five samples: pristine NVPF, NVPF-2.5, NVPF-3.0 discharged to 3.0 and 1.0 V, and NVPF-2.0 discharged to 1.0 V (Fig. 6). The shifts of the NMR resonance in these samples are Fermi contact shifts, which results from the presence of unpaired electrons on $V^{3+}$ or $V^{4+}$ ions. Therefore, $^{31}$P NMR probes the state of the four neighboring vanadium ions, while $^{23}$Na NMR provides a signature of the oxidation states of the two adjacent vanadium ions although sodium mobility and electron hoping may affect the observed spectra. Upon charging, the "narrow" $^{23}$Na and $^{31}$P lines (surrounded by $V^{3+}$ ions in pristine NVPF, Fig. 6a, f) have reduced shifts upon removal of vanadium electrons (Fig. 6e, j)[24,25]. A detailed analysis is provided in the Supplementary Note 1. The main conclusion that can be drawn from the NMR analysis results from the comparison of the spectra of pristine NVPF and NVPF-2.0 discharged to 1 V (i.e. Fig. 6a, f, e, j), and of NVPF-3.0 discharged to 3 and 1 V (Fig. 6c, d, h, i). On the one hand, it clearly confirms, as observed before, that NVPF-2.0 returns to its original ordered state upon discharge. On the other hand, for NVPF-3.0 upon discharge, the broad distributions of environments in $^{23}$Na and $^{31}$P spectra results from a distribution of vanadium oxidation states associated with sodium site occupancies, which are responsible for the variety of Fermi contact shifts on the neighboring phosphate groups and sodium ions, thereby confirming the nature of the observed disorder. Altogether, NMR results unambiguously confirm the presence of sodium disorder in the $Na_3V_2(PO_4)_2F_3$ phase formed by initial removal of three sodium ions.

**Charge compensation mechanism upon three sodium ions extraction/insertion**. To recap the overall picture of the structural changes involved here, a tetragonal phase of approximate composition $Na_0V_2(PO_4)_2F_3$ is formed when more than 2.5 sodium ions are extracted. Upon discharge, this phase accommodates sodium in a disordered way and does not convert back to the initial structure, further confirming that upon oxidation we prepared a new type of NVPF. This phase can uptake reversibly nearly three sodium ions with limited volume change ($\Delta V/V$ = 3.0%) as deduced by in situ XRD (Supplementary Figure 9) hence accounting for the good cyclability of the material. Thus, a question that arises concerns now is the origin of this disorder. Equally important is the charge compensation mechanism occurring through this reversible process, which is the evolution of the V oxidation state that we could not access directly by XRD as it only gives hints through V–O and V–F distances.

To get further insights into the evolution of the V oxidation state upon Na extraction-insertion, X-ray absorption spectra (XAS) was measured at the V L-edge and O K-edge for ex situ samples at Na contents referred by an asterisk on the voltage-composition curve (Fig. 7a). The normalized spectra acquired in total fluorescence yield (TFY) mode are presented in Fig. 7b, c. The V L-edges for the pristine NVPF show two peaks at 517.5 and 523.5 eV, which can be ascribed to the L2/L3 splitting with the additional multiplet effect originated from large 2$p$-3$d$ and 3$d$-3$d$ Coulomb and exchange interactions, as suggested by Abbate et al. for $V_2O_3$ and $V_2O_5$[26]. During charge, the existing peaks progressively shift to higher energies from 524.0 to 524.5 eV for instance, indicative of the vanadium oxidation since we observed a similar evolution when comparing with reference spectra taken on $V_2O_3$, $Na_1V_2(PO_4)_3$ and $V_2O_5$ having V at the 3+, 4+, and 5+ oxidation states, respectively (Supplementary Figure 10). These signals shift back during discharge, showing the reduction of vanadium back to the original valence, hence further confirming a reversible process. To gain further insight into the changes in electronic structure, we turn to the O K-edge reported

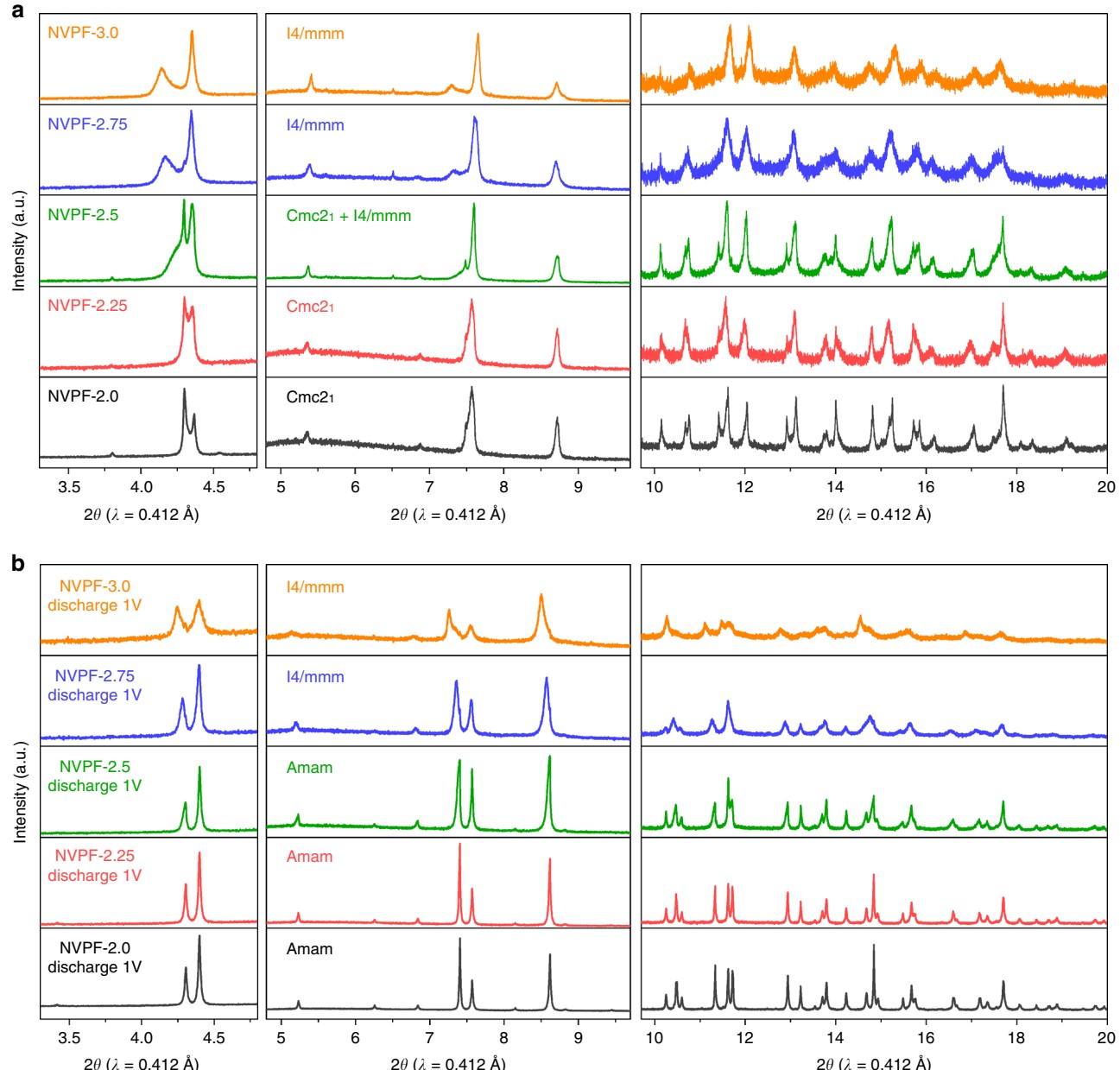

**Fig. 4** Structural evolutions during three sodium ions extraction/insertion in NVPF. **a** Synchrotron X-ray diffraction patterns of the charged samples with different amounts of Na extracted (NVPF-2.0, NVPF-2.25, NVPF-2.5, NVPF-2.75, and NVPF-3.0). **b** The other set of samples further discharged down to 1 V

in Fig. 7c, where the most relevant feature lies in the pre-edge signal. It is associated to the transition from O(1s) to the antibonding V(3d)–O (2p) hybridized levels and the intensity is related to the V–O covalence[26]. Since V–O covalence increases with the oxidation state of vanadium, the intensity of the pre-edge indirectly gives information on the valence of the vanadium atoms. The pre-edge signal shows a clear intensity increase/decrease upon sodium ion removal/uptake, indirectly confirming the oxidation/reduction of vanadium, through the whole process between 4.8 and 1 V.

While we demonstrated a change in V valence, we noted the appearance of a well-defined peak at the O K-edge consistent with the emergence of a single valence and environment of V on

charging. This observation is consistent with a recent synchrotron XRD studies[21] reporting the disproportionation of $V^{4+}$ into equal amounts of $V^{3+}$ and $V^{5+}$ in $Na_1V_2(PO_4)_2F_3$. By fitting the pre-edge signals of the O K-edge collected for our $Na_1V_2(PO_4)_2F_3$ with four Gaussian peaks at 529.3, 531.5, 531.9, and 533.3 eV, we form two sets of doublets corresponding to $V^{5+}$ (529.3–531.9 eV) and $V^{3+}$ (531.5–533.3 eV). This treatment was extended to all samples and we plotted the variation of the $V^{5+}/V^{3+}$ signal ratio upon Na extraction (see Supplementary Figure 11 and Supplementary Tables 8, 9). It continuously increases upon extraction suggesting that the $V^{4+}$ disproportionation happens all along the extraction process with the opposite variation occurring during discharge, except from a small shift of the $V^{5+}$ $t_{2g}$ state. This shift

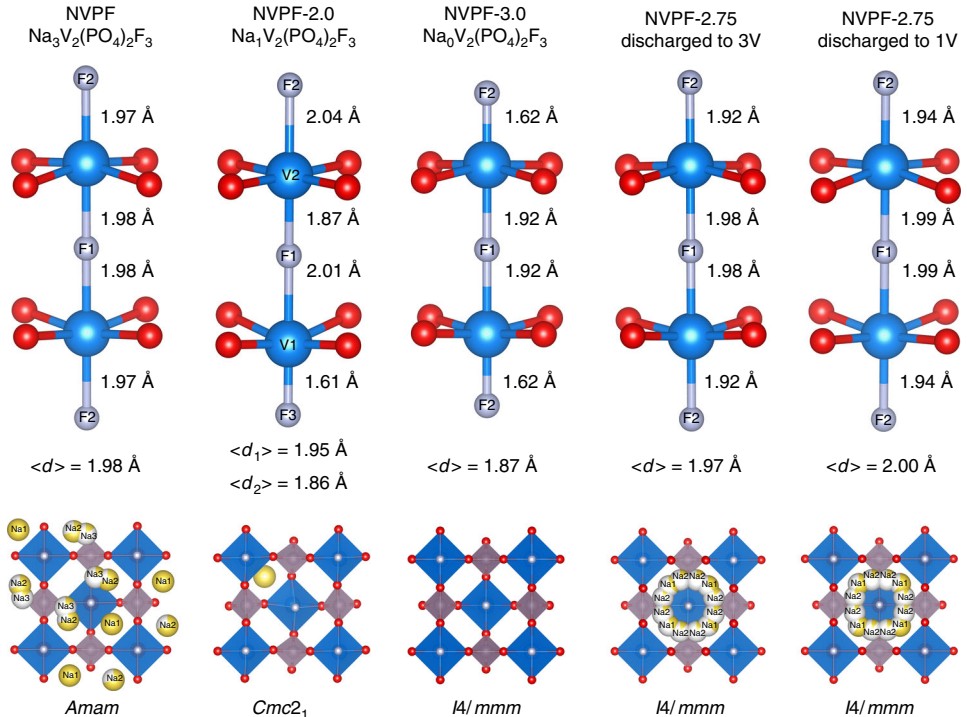

**Fig. 5** V–F bond lengths and sodium distribution within the structure of NVPF samples. Structures of NVPF samples, from left to right: pristine, NVPF-2.0, NVPF-3.0, NVPF-2.75 discharged to 3.0 V, and NVPF-2.75 discharged to 1.0 V. Top: view of the $VO_8F_3$ bioctahedra with relevant V–F bond lengths and the average V–O/F distances labeled as $<d>$; bottom: distribution of Na atoms within the structure. For each composition, the space group is indicated. V is blue, O is red, F is gray, Na is yellow, and Na vacancies are milky white

can be attributed to an increase in the number of sodium

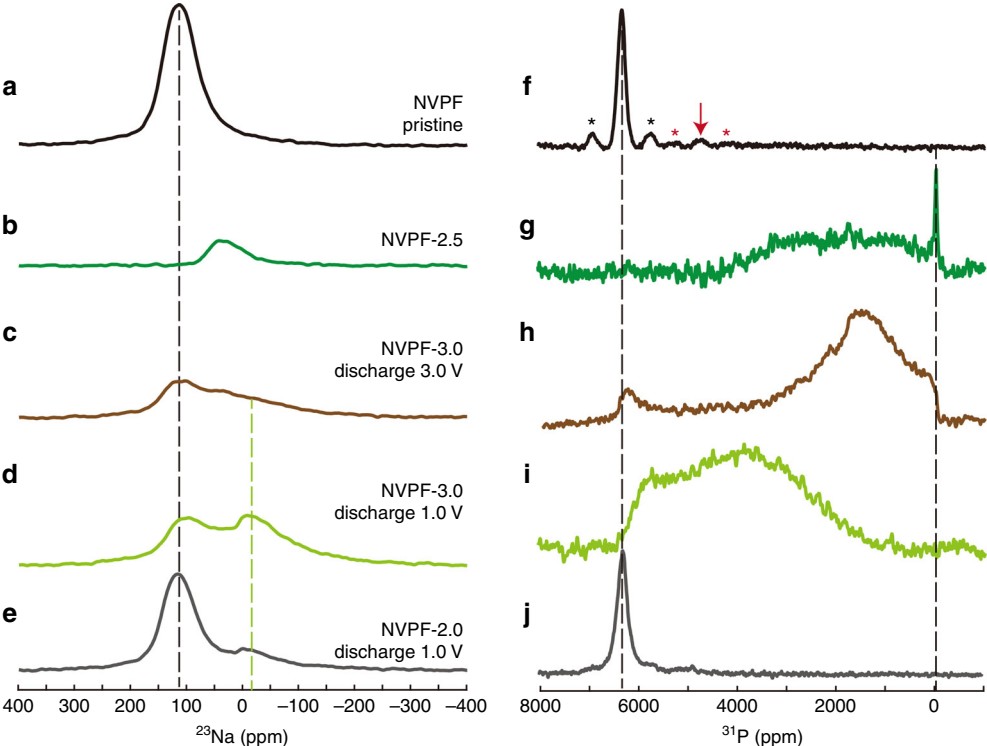

**Fig. 6** NMR spectroscopic evidence of sodium disorder in NVPF. **a–e** $^{23}Na$ and **f–j** $^{31}P$ magic angle spinning-nuclear magnetic resonance spectra after a Hahn echo sequence recorded at 4.7 T and a spinning rate of 50 kHz. **a**, **f** Pristine NVPF. **b**, **g** NVPF-2.5. **c**, **h** NVPF-3.0 discharged to 3.0 V. **d**, **i** NVPF-3.0 discharged to 3.0 V and 1.0 V respectively. **e**, **j** NVPF-2.0 discharged to 1.0 V. The stars indicate spinning sidebands, the red arrow shows a small contribution from neighboring $V^{4+}$ ions in the pristine NVPF. The dashed lines are guide for the eyes. The $^{23}Na$ spectra are shown without modification, while the $^{31}P$ spectra levels were adjusted for clarity

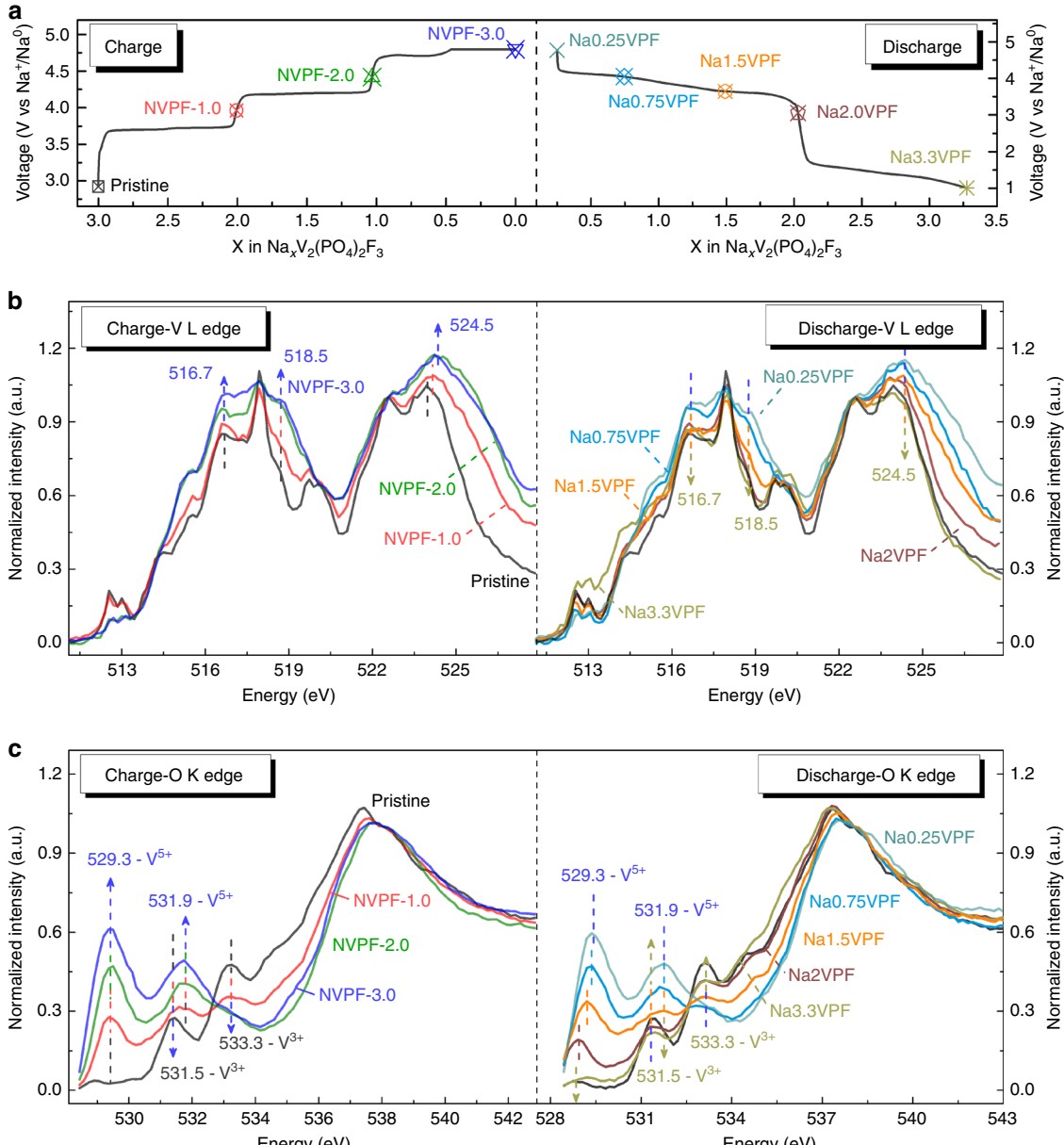

**Fig. 7** Evolution of the vanadium oxidation state upon Na$^+$ extraction/insertion. **a** The voltage-composition curve of NVPF-3.0 sample cycled at C/10 in which the experimental points used for ex situ X-ray absorption analysis are marked with the respective Na-stoichiometry. The normalized (**b**) V L-edge and **c** O K-edge X-ray absorption spectra of the corresponding charge (left) and discharge (right) samples obtained in total fluorescence yield mode

neighbors, which will change the Madelung energy and shift the V$^{5+}$ peak, in agreement with XRD data.

**Electrochemical performance of NVPF/C full cells by utilizing the third sodium in NVPF.** Next, we have implemented the aforementioned fundamental/experimental knowledge to the assembly of optimized NVPF/C cells. Figure 8 illustrates the performances of full cells, having a mass ratio positive to negative electrode of 2, and using an increasing fraction of the third plateau capacity during the first cycle. The cells cycled between 4.3 and 2 V show an increase in capacity from 107 to 121 mAh g$^{-1}$ for the NVPF-2.5 as compared to NVPF-2.0 as shown in Fig. 8a. The extra capacity gained by utilizing about half of this new third plateau translates into a ~14% increase in energy (396–451 Wh kg$^{-1}$ based

on the mass of NVPF), while preserving excellent capacity retention (Fig. 8b). Within this context, it is worth recalling that the highest achievable energy will depend on the value of the mass ratio of positive to negative ($r$), the voltage scanning domain, and the amount of sodium removed on the third plateau during the first charge. For instance, the maximum benefits in terms of capacity and energy when cycling over the 4.3–2 V voltage range is obtained with NVPF-2.6 sample using a NVPF/C mass ratio of 1.98 (Supplementary Figure 12), which is for instance 2% higher than what can be achieved with NVPF-2.5 using a mass ratio of 2.06 (Supplementary Figure 12). Note also that we can recover the signature of the low voltage plateau pertaining to the NVPF positive electrode by lowering the full cell voltage to zero volt. In that case, with $r = 2$, the increase in energy density approaches ~18% for NVPF-2.75 with respect to NVPF-2, but it must be recognized that some of the

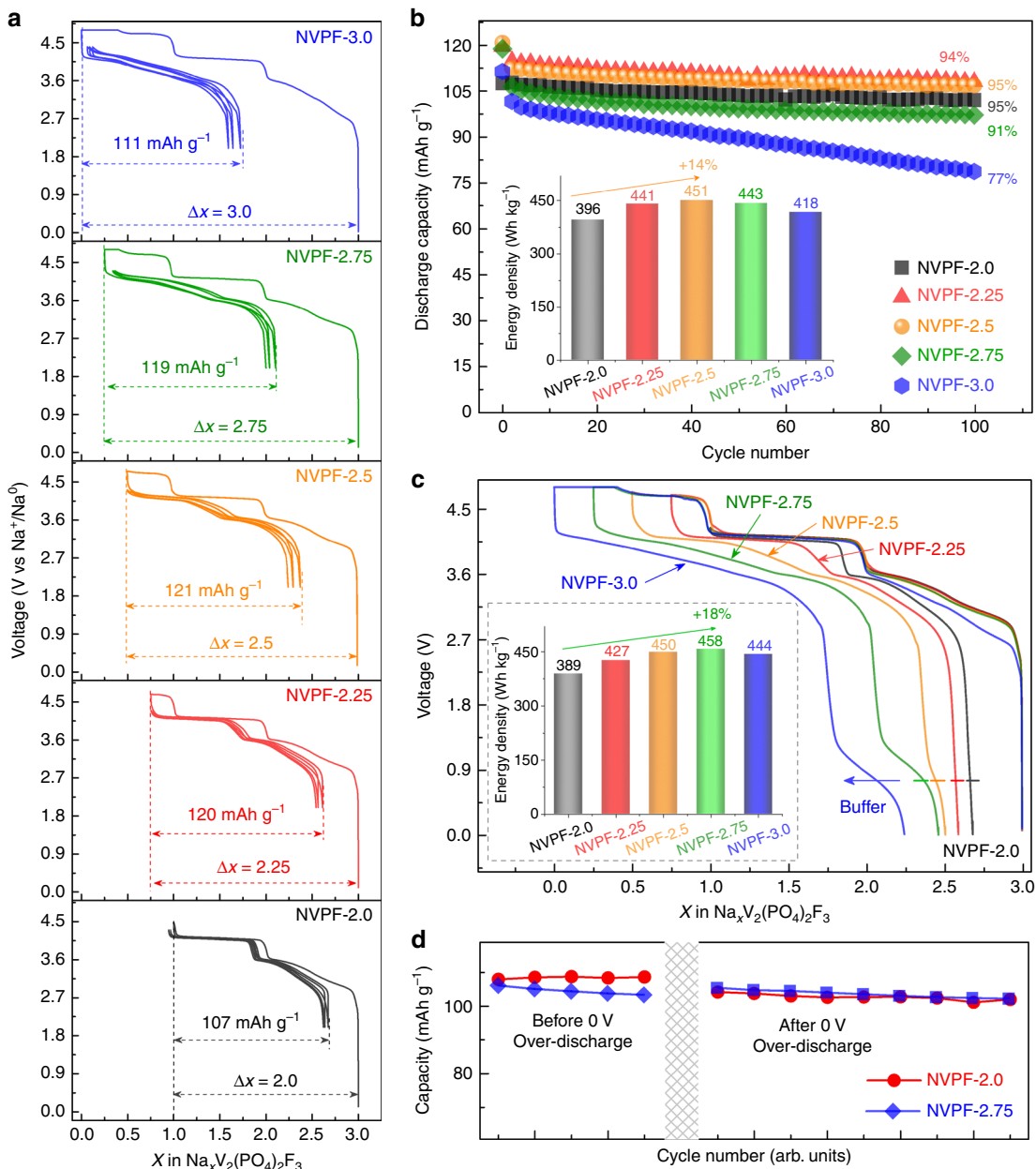

**Fig. 8** Electrochemical performance of NVPF/C full cells. The voltage-composition curves **a** and cycling performance **b** of NVPF/C full cell of NVPF-2.0, 2.25, 2.5, 2.75, and 3.0 samples cycled between 4.3 and 2.0 V after first activation process up to 4.8 V at C/10 followed a potentiostatic charging until reaching the exact amount of Na$^+$ content, where insets in **b** is the histogram of the discharge energy density based on the mass of NVPF. **c** The voltage-composition curves of NVPF/C full cell of NVPF-2.0, 2.25, 2.5, 2.75, and 3.0 sample cycled down to 0 V with the inset showing the histogram of energy density based on the mass of NVPF. **d** The cycling performance of NVPF-2.0 and NVPF-2.75 in a full cell before and after zero volt over-discharge tests for 1 week

extra-gain comes from the low-voltage plateau, which application-wise is not the most attractive. Altogether, these results unambiguously prove the energy density benefits of partially or wholly harvesting the third sodium ion from NVPF.

Lastly and worth stressing is the low-voltage plateau occurring at 1.6 V in NVPF/Na cell, which is shifted to 0.9 V in NVPF/C full cells, with its amplitude increasing as $\Delta x$ increases. To fully explore this low-voltage domain, two cells were previously charged to $\Delta x = 2$ and 2.75, and then cycled five times between 4.3 and 2.0 V first, and maintained at zero volts for 1 week prior

to be recharged and cycled again over the 4.3–2 V voltage domain. Note that the NVPF-2 loses 4% of its capacity during the 0 V resting period, while no capacity lost is observed for NVPF-2.75 sample (Fig. 8d and Supplementary Figure 13). Most likely, the low-voltage plateau serves as a buffer to clamp the voltage of both the NVPF positive and carbon negative electrodes to lower values so as to minimize side reactions during the over-discharge tests. From a practical perspective, this low-voltage plateau is an inherent asset for facilitating the handling (transport and storage of cells at zero volts) of NVPF/C sodium ion cells without

prejudice in their following uses, which is not the case with the Li-ion technology for instance.

## Discussion

The results reported here introduce the feasibility to reversibly remove, in contrast to previous beliefs, the third sodium ion from NVPF, hence offering this material and the resulting NVPF/C ion cells an additional asset for practical applications in terms of energy density and ability to be stored at 0 V without a performance penalty. Fundamental studies from complementary XRD, XAS, and electrochemical experiments establish that this extraction is associated with the oxidation of $V^{n+}$ beyond 4+ and is concomitant with the formation, structurally wise, of a new tetragonal "NVPF" phase, which can reversibly uptake three sodium ions; two and one at high and low potentials, respectively. However, as it is often the case with new results in the area of energy storage, the fundamental science still needs to be understood. Specifically, what are the mechanisms for sodium ion extraction/uptake, sodium ion transport, and structural reacting paths? Some of these points are now discussed in light of our experimental findings.

Classical extraction/insertion processes usually lead to charge and discharge curves, which mirror each other. Such a condition is no longer satisfied here since the first discharge profile differs drastically from the charge one, once the third sodium is extracted from the NVPF structure. This leads to the irreversible formation of new "NVPF" disordered compound as deduced by XRD, which remains as such through subsequent cycling. Such a situation is not unique and has been observed in $Li_2FeSiO_4$ and $Li_3VO_4$, where it was ascribed to the onset of a Li/M interstice mixing upon reduction[27,28]. Such a scenario is quite unlikely here owing to the large size difference between $Na^+$ and $V^{5+}$. Another possibility would be a strengthening of the $V=O$ bond at the expense of a lengthening of the V–F bond toward the end of the oxidation, making it more vulnerable toward bond breaking. This could lead to the formation of F vacancies having the possibility to permanently modify Na distribution. However, such a hypothesis could not be validated owing to our inability to (i) reach satisfactory Rietveld refinements upon introduction of F vacancies and (ii) to detect fluorine by $^{19}F$ NMR in the electrolyte recovered from a cell that was fully charged using $NaClO_4$ rather than $NaPF_6$-based electrolyte. Most likely, under strong oxidation conditions, we change the energy of the Na vacancy landscape via some subtle framework modifications, which cannot be clearly determined with powder XRD. This could promote permanently a more energetically favorable disordered Na path that, once formed, remains on subsequent cycles.

Turning to the evolution of the vanadium oxidation state upon $Na^+$ extraction/insertion, our XAS data together with bond valence sums deduced from XRD indicate that vanadium can reach an average oxidation state approaching 4.5+. However, ambiguity remains as whether oxidation states ($V^{3+}$, $V^{4+}$, and $V^{5+}$) coexist within the same material. Their coexistence upon sodium extraction was shown in $Na_3V_2(PO_4)_2O_{1.6}F_{1.4}$ via X-ray absorption near edge structure spectroscopy[29]. In contrast, recent synchrotron XRD studies, which we confirmed by soft XAS, support the existence of V disproportionation ($2V^{4+} \rightarrow V^{3+} + V^{5+}$) in $Na_1V_2(PO_4)_2F_3$ that is associated with the instability of $V^{4+}$ in $VO_4F_2$ octahedrons[23,25,30]. These observations indicate the feasibility of having $V^{4+}$ in O-substituted NVPF, but not in pure NVPF compounds having $VO_{5-x}F_x$ and $VO_4F_2$ octahedron. Let us recall that within the NVPF-type structure, V ions sit in the center of octahedra linked by vertices through fluorine atoms to form $V_2O_8F_3$ bioctahedra, while in the O-substituted phases, $V_2O_{8+x}F_{3-x}$ bioctahedra present V–O–V bonds along the $c$-axis. Obviously, the large predominance of V–F–V bonds in contrast to V–O–V bonds

will limit the possibility of stabilizing vanadyl groups. We thus believe that the difficulty in stabilizing $V^{4+}$ in the fluoride environment provided by $VO_4F_2$ octahedron favors the disproportionation of $V^{4+}$ into $V^{3+}$ and $V^{5+}$.

Having consolidated, based on complementary GITT and galvanostatic measurements, that the origin of the potential jump is not kinetic in nature, a question remains regarding the onset of this potential jump, which occurs at $x = 2$ in our newly disordered $Na_xV_2(PO_4)_2F_3$ phase instead of $x = 3$ in NVPF or O-substituted NVPF. A tentative answer can be deduced by considering the phase that forms at high potential during charge, which influences the discharge profile with the amount of sodium release at high potential being nearly equal to that reinserted at low potential. Let's note that the $I4/mmm$ structure contains two different Na sites, A and B sites, that could be the Na2 and Na1 on Wyckoff positions $16l$ and $8h$, respectively (Supplementary Tables 4 and 5). One can simply imagine that, first, the A sites depopulate ($Na_3V_2(PO_4)_2F_3 \rightarrow Na_1V_2(PO_4)_2F_3$) followed by the B sites at high potential, and then A sites refill prior to the B sites so that the sodium ion liberated from B sites at ~4.7 V are solely reinserted at ~1.6 V. Such a scenario leads to different metastable intermediate states in agreement with the different thermodynamic reaction paths observed via GITT measurements. Moreover, it could then explain the need to have two reinserted sodium ions corresponding to A sites, whatever the amount of sodium ions extracted in charge, before the potential jump occurs. The large potential difference (~4.7 − 1.6 = 3.1 V) between the depopulation/repopulation of B sites could be explained by different sodium environments associated to different $V^{(n+1)+}/V^{n+}$ redox couples. Further exploration of the sodium-driven local structural changes in NVPF by combined transmission electron microscopy, electron paramagnetic resonance, and operando XAS studies are needed to further support this scenario, since neither lab-XRD nor synchrotron sources can sort out differences between sites occupancies.

Overall, this study contradicts the conventional understanding of the electrochemical properties of $Na_3V_2(PO_4)_2F_3$, by showing that the third sodium ion can be reversibly removed, which leads to a new polymorph having disordered rather than ordered sodium sites, and a tetragonal symmetry instead of an orthorhombic one. We found that the amount of sodium ions removed mirrors the amount reinserted at low potential and explained this balance by the existence of two thermodynamic paths corresponding to different sites occupancies sequences. Furthermore, this report offers a new way to enhance the energy density of $Na_3V_2(PO_4)_2F_3$ batteries by 14% while preserving excellent cycle life and suitable rate capabilities for applications. Lastly, we demonstrate that the onset of the low insertion plateau constitutes a practical step toward the design of sodium ion cells having performances unaffected by maintaining or discharging the cell to zero volts. Hence, these novel insights should help in boosting the development of the sodium ion technology.

## Methods

**Electrochemical characterization**. Electrodes were mainly consisting either of 11 mm diameter disks punch out of Al supported calendared NVPF/Csp/PVDF tapes with a 92/4/4 in weight ratio and a loading of 12 mg cm$^{-2}$. Powdered composite mixtures (NVPF/Csp with a 90/10 in weight ratio) were occasionally used. A unit of 1 M NaPF$_6$ (Stella, Japan) dissolved in PC (BASF, Germany) was used as the electrolyte, glass fiber (GF/D, Whatman) was used as separator, and sodium metal (Sigma-Aldrich) and hard carbon were used as negative electrode for half cells and full cells respectively, throughout the paper unless otherwise specified. In the initial stages of the exploration of the third plateau of NVPF (Supplementary Figure 1), the electrolyte 1 M NaPF$_6$ dissolved in PC/EC/DMC (1/1/1 in volume ratio) was used. The Swagelok and coin-type cell were assembled in the glovebox (MBRAUN, Germany) either in half cells or full cells, in which the current density of C/10 (1C = 128 mA g$^{-1}$) was applied by a MPG-2 or VMP-3 potentiostat (Bio-Logic, France). The potentiostatic mode by limiting the current to less than C/100 at 4.8 V (vs. Na$^+$/Na$^0$) was only

employed in the first formation charge process. The GITT (every $\Delta x = 0.1$) was performed in the second cycle of NVPF-2.75 sample after the first activation cycle, and the relaxing process was controlled either by $dV/dt \leq 0.1$ mV s$^{-1}$ or 4 h.

**Ex situ synchrotron XRD.** First, the designed amount of sodium ions were extracted from or reinserted into the NVPF structure electrochemically in the Swagelok cell, and the recovered powder was washed with dimethyl carbonate (DMC) and dried in vacuum before sealing it in 0.7 mm glass capillaries for synchrotron XRD measurements. The ex situ synchrotron XRD measurements were performed using the mail in user facility at 11BM synchrotron beamline, Argonne National Laboratory ($\lambda = 0.412$ Å). All Rietveld refinements were performed using the FullProf program[31].

**Ex situ soft X-ray absorption.** A series of charged and discharged samples (NVPF electrodes here) with certain amount of sodium ions extracted were prepared in the Swagelok type cell, and the cycled samples are washed, dried, and sealed in argon for XAS measurements before sealing in the bags made of Al-plastic film. Samples were then mounted in a glovebox and transferred under Argon environment into the beamline end station. The L-edge of V and O K-edge XAS spectra was obtained in both surface (total electron yield) and bulk sensitive (TFY) simultaneously at beamline 4-ID-C of the Advanced Photon Source. All spectra were aligned by the simultaneous measurement of an MgO reference sample.

**Magic angle spinning-nuclear magnetic resonance.** The spectra were recorded on a Bruker 4.7T (200 MHz for $^1$H) double resonance spectrometer, operating at 53 MHz for $^{23}$Na and 81 MHz for $^{31}$P, using a 1.3 mm double resonance magic angle spinning probe, with a spinning rate of 50 kHz, and N$_2$ for the bearing, drive, and frame cooling gas flows. The 1.3 mm zirconia rotor were filled in a glovebox under Argon. The effective RF field strength was set to 250 kHz for $^{23}$Na and $^{31}$P, using a Hahn echo sequence over two rotor periods (i.e. 40 µs) to obtain a distortion-free baseline. In the case of $^{31}$P, the spectra were recorded using variable offset-cumulative spectrum—every 250 kHz—to record the complete signal[32]. The spectra were fitted with DmFit for the quantification of each peak when needed, using standard Gausso-Lorentzian peaks[33]. The spectra were referenced with a 1 M NaCl solution in water for $^{23}$Na and 85% H$_3$PO$_4$ in water for $^{31}$P. The longitudinal relaxation times $T_1$'s were around 0.5 to a couple of milliseconds, and recycling delays of 30 ms were used to ensure a complete recovery of the magnetization. For $^{23}$Na and $^{31}$P, 4096 and 16,384 transients were recorded for each spectrum/offset, and exponential line broadening of 500 and 2000 Hz were applied, respectively.

## Data availability
The data supporting the findings of this study are available from the authors on reasonable request.

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

## Acknowledgements
J.-M.T. acknowledges funding from the European Research Council (ERC) (FP/2014)/ERC, Grant-Project 670116-ARPEMA; Q.J. thanks the ANR "Deli-Redox" for Ph.D. funding. This research used resources of the Advanced Photon Source (11BM and 4-ID-C), a U.S. Department of Energy (DOE) Office of Science User Facility operated for the DOE Office of Science by Argonne National Laboratory under Contract No. DE-AC02-06CH11357. The authors thank M.-L. Doublet and C. Delacourt for fruitful discussion together with A. Abakumov and J. Hadermann for looking at some of the reported samples by TEM, and D. Giaume for ICP analysis. M.D. acknowledges support from CNRS and LabeX STORE-EX for funding.

## Author contributions
G.Y., J.-M.T. and S.M. designed and conducted the experimental work. G.R. analyzed the crystal structure of all samples. M.D. performed the MAS-NMR experiments and analyzed the data. R.D. synthesized the pristine material of Na$_3$V$_2$(PO$_4$)$_2$F$_3$. Q.J. and J.F. participate in the acquisition and analysis of the soft X-ray absorption data. B.M. conducted the elemental analysis by EDX. All authors discussed the results and contributed to draft the manuscript.

## Additional information

**Competing interests:** The authors declare no competing interests.

