## [Peer Review File · Nature Communications]

Reviewers' comments:

Reviewer #1 (Remarks to the Author):

In this work, Tarascon and coworkers present the evidence of the feasibility of the extraction of the third Na from $\text{Na}_3\text{V}_2(\text{PO}_4)_2\text{F}_3$ phase. Moreover, the extraction of this third sodium induces changes in the structure that change the electrochemical signature of this compound as cathode for Na-ion batteries. These findings are of great interest to the community, are well arranged and explained. There are only some minor issues to be clarified through the text to allow a better understanding of it. The issues are the following ones:

- when comments are made on XAS measurements, line 201, it is said that "new signals progressively appear at higher energies...", I cannot see new signals in figure 6 but a shift on the existing ones to higher energies, this should be clarified in the text.
- when explaining the XAS of the O k-edge (line 208-209), it is said "It is associated to the transition from O(1s) to the anti bonding V(3d-O82p) hybridized levels and the intensity is related to the V-O covalence". I think that this statement should be explained more thoroughly to clarify the relationship between V-O covalence and the oxidation state of vanadium.
- also some references regarding overcharge of sodium vanadium fluorophosphates up to 4.8 V should be added, because there exist literature on this issue. Although none of them maintained this voltage during some time, perhaps it is worth commenting on the differences in the structure when 4.8 V are reached or when this high voltage is maintained.

Reviewer #2 (Remarks to the Author):

This manuscript reported an in-situ electrochemically made disordered $\text{Na}_3\text{V}_2(\text{PO}_4)_2\text{F}_3$ (NVPF) cathode materials in sodium ion batteries. The main novelty of this work lies in making it possible to use the 3rd Na in NVPF via the formation of a disordered phase. The Na-driven phase and structural evolutions were systematically investigated by X-ray diffraction and absorption measurements. In addition, the cyclic capacities and energy densities of NPVF half cells and carbon/NVPF full cells were moderately improved. Although this work presented some interesting findings, several critical concerns discourage me to recommend its publication in Nature Communications:

1. Although the disordered NVPF materials enable the reversible insertion of the 3rd Na, its working potential is very low (~1.6 V), which does not present any practical merits in the sodium full cell.
2. In the section of studying NVPF materials, the authors claimed that NVPF-3 presents the highest capacity by using the 3rd Na. However, in the section of NVPF/C full cells, the author demonstrated NVPF-2.35 presented the best performance other than NVPF-2.75 or NVPF-3. It suggests that the 3rd NVPF-3 cannot even be fully utilized in full cells, which discourages the contribution of this work.
3. Material characterization methods used in this work are limited to XRD and XAS, which integral techniques are insufficient to claim the structures of materials. Nevertheless, significant lack of local structural characterization is problematic to support the formation of the new disordered structure. More detailed probes of NVPF-2, NVPF-2.25, NVPF-2.5, NVPF-2.75 and NVPF-3 are needed to show their crystal structures and chemical compositions.
4. In Page 8, the author stated: "the disordered reduced phases present the same symmetry and crystal structures (Na distribution) as the high temperature form of $\text{Na}_3\text{V}_2(\text{PO}_4)_2\text{F}_3$ ". It is suggested to directly prepare the high temperature phase to compare their electrochemical properties and to verify their claims of utilizing the 3rd Na in their "new $\text{Na}_3\text{V}_2(\text{PO}_4)_2\text{F}_3$ material". In addition, the authors are suggested to compare the electrochemical performance of current NVPF-3 with its peers to evaluate the standing among literatures.
6. Some typo:

In Page 3: “reversible electrochemical insertion of 0.5 Na⁺ in Na₃V₂(PO₄)₂F₃ or 1 Na⁺ in Na₃V₂(PO₄)₂O₂F at potentials of 1.4 and 1.5 V, respectively” should be “ ≈ 1.6 V”.

In Page 20: “and the long cycling performance of these NVPF/Na cells cycled between 4.4 to 1 V at C/10 (c).” 25 cycles are not long.

Reviewer #3 (Remarks to the Author):

In this manuscript, G Yan et al. studied the correlation between structure and electrochemistry of highly delithiated Na₃(VPO₄)₂F₃ electrode (NVPF) in Na⁺ batteries. Specifically, the authors were able to extract the third Li from NVPF relying on their previous electrolyte engineering results. The totally delithiated NVPF shows a higher symmetry than pristine NVPF. After relithiation, the same higher symmetry structure was maintained, in contrast to the case when Li is not totally removed. The higher symmetry structure shows a strong resemblance of NVPF at higher temperature (a previous study). Furthermore, the electrochemical effect of fully delithiation, in both half cells and full cells, was demonstrated. Relithiation of the 3rd Li leads to a low voltage plateau, and will moderately improve the overall performance of NVPF. Also, additional evidence on charge compensation mechanism was provided (XAS studies). This manuscript reports a quite interesting phenomenon about the content of lithiation leading to significant structural change and battery behavior of a material, which is worth readership in this community. At this stage, there are a few concerns about the details of the paper.

1. In the first place, as far as I am concerned, the key to extract the 3rd Li by modifying the electrolyte is a little confusing. In Figure S1 part, it was stated that: “Thus, by further increasing the oxidation potential beyond 4.2 V using DMC containing electrolyte showed appearance of prolonged plateaus at ~ 4.7 V (Fig. S1b).” But in the main text experimental section, it was described that the electrolyte solvent is pure PC. This part should be more clarified in order to avoid confusion for readership.

2. The terminology “disordered” may not be totally appropriate in this context. I understand the terminology was started by ref. 20 (Chem Mater 430 26, 4238-4247 (2014).). The Li sites just have higher multiplicity (8 and 16) and low occupation. On the basis of refinement results, Li is not sharing any sites with other cations (not like cation mixing between Li and Ni in NMCs and disordered cathodes). The application of this terminology should be considered (even previous publication has used this terminology).

3. For the comparison of half cell and full cell behavior. In Figure 2 (half cell test), NVPF-3.0 is the most different one (comparing to partially delithiated NVPFs). However, why is full cell behavior of NVPF-3.0 not mentioned?

4. Similarly, why is the number of cycle for NVPF-3.0 is much fewer than others in Figure 2 (half cell), and the number of cycle for NVPF-2.75 is much fewer than others in Figure 7 (full cell). NVPF 2.75 and 3 are the highlights of the paper, so I believe the number of compared cycles should be same with others.

5. In line 175, page 8, it was stated that: “Interestingly, the $\Delta x = 2.75$ and $\Delta x = 3$ samples stand as an exception, since they never convert back into the pristine Amam phase as shown in Fig. 4b.”. However, in Fig 4b, the XRD of $\Delta x = 3$ sample was not shown.

6. The authors applied soft XAS V-L edge and O-K edge to explain the V charge compensation mechanism. It is well known that soft-XAS strongly depends on surface properties and the results are strongly controlled by surface behavior. As far as I am concerned, hXAS should be a more accurate description of charge compensation mechanism.

Reviewer #1

- a) **When comments are made on XAS measurements, line 201, it is said that "new signals progressively appear at higher energies..." I cannot see new signals in figure 6 but a shift on the existing ones to higher energies, this should be clarified in the text.**

The referee is right, and we have modified the text accordingly. "During charge, the existing peaks progressively shift to higher energies from 524 eV to 524.5 eV for instance, indicative of the vanadium oxidation". Please see page 8, line 194- 195 in the revised manuscript.

- b) **When explaining the XAS of the O k-edge (line 208-209), it is said "It is associated to the transition from O(1s) to the anti bonding V(3d-O2p) hybridized levels and the intensity is related to the V-O covalence". I think that this statement should be explained more thoroughly to clarify the relationship between V-O covalence and the oxidation state of vanadium.**

We propose to add: It is associated to the transition from O(1s) to the antibonding V(3d)-O (2p) hybridized levels and the intensity is related to the V-O covalence. Since V-O covalence increases with the oxidation state of

vanadium, the intensity of the pre-edge indirectly gives information on the valence of the vanadium atoms. The pre-edge signal shows a clear intensity increase/decrease upon Na⁺ removal/uptake, indirectly confirming the oxidation/reduction of vanadium, through the whole process between 4.8 V and 1 V. Please see page 8, line 201-203.

- c) **Also some references regarding overcharge of sodium vanadium fluorophosphates up to 4.8 V should be added, because there exist literature on this issue. Although none of them maintained this voltage during some time, perhaps it is worth commenting on the differences in the structure when 4.8 V are reached or when this high voltage is maintained.**

This question bothers us at first as we carefully look at the literature and do not find any paper on the overcharge of NVPF. In contrast, we found a paper entitled “Structural evolution of mixed valent (V³⁺/V⁴⁺) and V⁴⁺ sodium vanadium fluorophosphates as cathodes in sodium-ion batteries: comparisons, overcharging and mid-term cycling by V.A Palomares et al (J. Mater. Chem. A, 2015, 3, 23017) that is dealing with the overcharge performance of the of Na₃V₂O_{1.6}(PO₄)₂F_{1.4} and Na₃V₂O₂(PO₄)₂F phases with the authors focusing on the possible degradation mechanisms (and phases) while overcharging. The goal of this paper is quite different as compared with ours, they have observed that there is disordered characteristic after the overcharge tests under 4.8 V for the Na₃V₂O_{1.6}(PO₄)₂F_{1.4} compound. Although it is a different system, we have added this reference in our revised manuscript.

Reviewer #2

- a) **Although the disordered NVPF materials enable the reversible insertion of the 3rd Na, its working potential is very low (~1.6 V), which does not present any practical merits in the sodium full cell.**

We agree with the point that the working potential of 3rd Na is quite low since we even wrote in the paper “Such a low potential limits the specific energy gain associated to the insertion of extra Na ions”. However, it does bring us some practical merits, such as 15% energy density increase when the 3rd Na is used to compensate for the irreversible Na-ions consumed by formation of the SEI at the carbon negative electrode in the NVPF/C full cell, although such a gain is far to be perfect in full Na-ion cells. Besides energy density, another figure of merit provides by the low voltage interaction of Na⁺ and stressed in this paper regarding safety, is its inherent asset for facilitating the handling (transport and storage of cells at zero volts) of NVPF/C Na-ion cells without prejudice in their following uses.

- b) **In the section of studying NVPF materials, the authors claimed that NVPF-3 presents the highest capacity by using the 3rd Na. However, in the section of NVPF/C full cells, the author demonstrated NVPF-2.35 presented the best performance other than NVPF-2.75 or NVPF-3. It suggests that the 3rd NVPF-3 cannot even be fully utilized in full cells, which discourages the contribution of this work.**

The referee is perfectly right and we anticipated this question since we could not include data for Δx=3 neither the plot below in our initial version because of pending patents issues (European (EP18305521.9)) that clear two days after submission. So as asked by the referee we modified our previous plots by injecting the voltage-composition-cycling curves for Δx = 3 as well the corresponding XRD's. Moreover, we added in supplementary information the plot below (Fig. S12) nicely conveying the practical aspects of increasing amount of Na⁺ removed from the 3rd plateau. This plot shows the variation of the cell energy density in full Na-ion coin cells cycled from 4.3 to 2 V as a function of key parameters such as i) mass ratio of + to – electrode, ii) the amount of removed Na

(Δx) while fixing the reversible capacity of the negative electrode at 250 mAh/g. It appears that the best composition capacity-wise is for $\Delta x=2.6$; a composition which falls into the Na removal range for which there is the formation of the disordered NVPF phase. Worth mentioning is that such maximum can vary depending upon the nature of the C electrode that controls the SEI., the greater the SEI formation, the greatest the shift towards $\Delta x=3$. Although capacitive-wise, there is no incentive to push the Na removal beyond 2.6 (except with poor carbons), an advantage of using $\Delta x > 2.5$ is to have Na-ion cells that can be easily and safely handled in their discharged state (0 V) while minimizing the penalty to be paid in terms of energy density.

Fig. S12 The variation of the cell energy (based on the total mass of NVPF and C) in full Na-ion coin cells cycled from 4.3 to 2 V as a function of key parameters of mass ratio of positive to negative electrode and the amount of sodium removal on first charge. The specific energy is calculated for the total mass of positive and negative active materials and the irreversibility on first cycle is mentioned for the weight of hard carbon electrode.

c) **Material characterization methods used in this work are limited to XRD and XAS, which integral techniques are insufficient to claim the structures of materials. Nevertheless, significant lack of local structural characterization is problematic to support the formation of the new disordered structure. More detailed probes of NVPF-2, NVPF-2.25, NVPF-2.5, NVPF-2.75 and NVPF-3 are needed to show their crystal structures and chemical compositions.**

We can indeed understand the referee doubt about the weakness of our experimental support regarding our disordered phase as we solely used **XRD and XAS** for structural characterization techniques. In contrast, we are strongly confident about the chemical composition of the phases we reported that were deduced from the cross-over between ICP and SEM-EDX results.

With respect to structural aspect, we first realized that we should have better define the type of disorder we were talking about **"It corresponds to Na distributed on Wyckoff sites of high multiplicity with an occupancy lower than 1, therefore it is not a chemical disorder/intersites mixing"**. Moreover, structural tables including lattice parameters and atomic positions are reported in the paper for NVPF-3 (Table S2), as well as a view of the Na distribution in that compound (Figure 4).

Whatever, the referee's question was an impetus to grasp further insight into this disorder and to study our compounds in light of what has been previously done on classical NVPF (Grey et al, Chem. Mater., 2014, 26 (8), pp 2513–2521). This explains i) our delay in providing this rebuttal document as well as ii) the addition of another co-author in the paper. **From the ^{23}Na and ^{31}P NMR spectra collected on 5 different samples we could**

clearly confirm the nature of the observed disorder in our newly made $\text{Na}_3\text{V}_2(\text{PO}_4)_2\text{F}_3$ polymorph. To convey this point we have added a 20 lines section (page 7, lines 162- 178) plus a figure in the main text (Fig. 6) together section devoted to NMR in the supplementary section (Supplementary note. S1). Again we are thankful to the referee for his/her question.

d) In Page 8, the author stated: “the disordered reduced phases present the same symmetry and crystal structures (Na distribution) as the high temperature form of $\text{Na}_3\text{V}_2(\text{PO}_4)_2\text{F}_3$ ”. It is suggested to directly prepare the high temperature phase to compare their electrochemical properties and to verify their claims of utilizing the 3rd Na in their “new $\text{Na}_3\text{V}_2(\text{PO}_4)_2\text{F}_3$ material”. In addition, the authors are suggested to compare the electrochemical performance of current NVPF-3 with its peers to evaluate the standing among literatures.

Interestingly, the high temperature form of NVPF (Bianchini paper 2014, as shown in Table R1) show distinct lattice parameters from ours (the two rows below). Therefore, we do not claim it is the same phase, but solely state that the structures have the same space group and Na distribution. They slightly differ in lattice parameters since the high T phase at 400K shows the same *a* as the NVPF-2.75 discharged to 3V but smaller *c*, and same *c* as NVPF2.75 discharged to 1V but smaller *a*.

Table R1. The structural parameters of High T NVPF (400 K) from Bianchini’s paper and our NVPF-2.75 samples

Redacted
Our data: Structural parameters of NVPF-2.75 discharged to 3 V determined from the Rietveld refinement of the synchrotron 11BM XRD data. $\text{Na}_x\text{V}_2(\text{PO}_4)_2\text{F}_3$ ($x = 2.40(1)$ from refinement), I4/mmm space group, $Z = 2$ $a = 6.39489(5) \text{ \AA}$, $c = 10.81238(14) \text{ \AA}$, $V = 442.17(1) \text{ \AA}^3$, $V/Z = 221.08 \text{ \AA}^3$, $R_{\text{Bragg}} = 6.6 \%$, $\chi^2 = 1.11$
Our data: Structural parameters of NVPF-2.75 discharged to 1 V determined from the Rietveld refinement of the synchrotron 11BM XRD data. $\text{Na}_x\text{V}_2(\text{PO}_4)_2\text{F}_3$ ($x = 3.25(1)$ from refinement), I4/mmm space group, $Z = 2$ $a = 6.43624(10) \text{ \AA}$, $c = 10.7602(2) \text{ \AA}$, $V = 445.74(2) \text{ \AA}^3$, $V/Z = 222.87 \text{ \AA}^3$, $R_{\text{Bragg}} = 7.55 \%$, $\chi^2 = 1.03$

We thank the referee for the suggestion that we had already tried. The high temperature phase reported by Bianchini et al was obtained by in situ XRD in temperature with the transition confirmed by DSC measurements. Therefore, this phase converts back to its initial ordered structure upon cooling, the reason why we tried to stabilize it at room temperature by quenching but without success. Moreover, we could not check its electrochemical behavior at 120°C because of the lack of suitable electrolytes for operating at such high temperature. So what we have done is to anneal our disorder phase ($\Delta x=2.75$ discharged to 3V) to various temperatures of 100, 150, and 200°C, so as to recover an ordered structure but we did not observe any structural

evolution (as shown in Fig. R1). Besides, there is no electrochemical evolution between the pristine and the annealed phase once cooled back to RT as shown in Fig. R2. So in short, we believe that the disorder 400K phase reported by Bianchini and our disordered phase are different.

Fig. R1. Structural parameters of NVPF-2.75 discharged to 1 V after heated at 200 °C determined from the Rietveld refinement of the synchrotron 11BM XRD data

Fig. R2. The charge and discharge curve of NVPF-2.75 sample before and after annealing at 200 °C

- e) A few mistakes in Page 3: “reversible electrochemical insertion of 0.5 Na⁺ in Na₃V₂(PO₄)₂F₃ or 1 Na⁺ in Na₃V₂(PO₄)₂O₂F at potentials of 1.4 and 1.5 V, respectively” should be “≈1.6 V”. In Page 20: “and the long cycling performance of these NVPF/Na cells cycled between 4.4 to 1 V at C/10” 25 cycles are not long.

We took care of all of these mistakes in our revised version, which has been corrected. Please see page 2, line 49, and page 18, caption of figure 2.

Reviewer #3

- a) In the first place, as far as I am concerned, the key to extract the 3rd Na by modifying the electrolyte is a little confusing. In Figure S1 part, it was stated that: “Thus, by further increasing the oxidation potential beyond 4.2 V using DMC containing electrolyte showed appearance of prolonged plateaus at ~4.7 V (Fig. S1b).” But in the main text experimental section, it was described that the electrolyte solvent is pure PC. This part should be more clarified in order to avoid confusion for readership.

We can understand this confusion as the DMC story is not described in detail. The only purpose of that sentence was to mention how we were brought to the study of the removal of the 3rd sodium. Indeed, trying to identify the best solvent electrolyte (Guochun et al, *Journal of The Electrochemical Society*, **165** (7) A1222-A1230 (2018)), we compared cyclic (EC, PC) vs. Linear (DMC, EMC, DEC) carbonates, thus we found DMC to be bad with the existence of an infinite plateau at 4.8 V. Therefore, once the current was switched on discharge the cell was able to reinsert 2 Na⁺ down to 3V via a smooth as opposed to stair case voltage profile. This was the first indication of a sodium deintercalation process affecting the structure of NVPF at high potential that was proceeding in parallel to electrolyte decomposition. Hence our decision to thoroughly investigate this point via the use of stable electrolytes (EC, PC, or their mixture). *We have slightly modified the text to clarify the use of electrolyte with DMC in the experimental section.* Please see page 13, line 314-316.

b) The terminology “disordered” may not be totally appropriate in this context. I understand the terminology was started by ref. 20 (Chem Mater 430 26, 4238-4247 (2014).). The Na sites just have higher multiplicity (8 and 16) and low occupation. On the basis of refinement results, Na is not sharing any sites with other cations (not like cation mixing between Li and Ni in NMCs and disordered cathodes). The application of this terminology should be considered (even previous publication has used this terminology).

Because this terminology was indeed already extensively used in literature, we prefer not to change the wording. However, for sake of clarity, we have redefined in the text what disordered is meant: “Note that disorder corresponds here to a distribution of sodium atoms among Wyckoff sites of high multiplicity (8 and 16), where they partially occupy (Na and vacancies on the same site).” Therefore, disorder does not refer to any cationic mixing. Please see page 7, line 154-156.

c) For the comparison of half cell and full cell behavior. In Figure 2 (half cell test), NVPF-3.0 is the most different one (comparing to partially delithiated NVPFs). However, why is full cell behavior of NVPF-3.0 not mentioned?

The absence of the data $\Delta x=3$ is a fully justified question that we answered to referee #2 (question b) by i) giving the reason for such an omission and ii) redrawing the figures with the data for $\Delta x = 3$.

d) Similarly, why is the number of cycle for NVPF-3.0 is much fewer than others in Figure 2 (half cell), and the number of cycle for NVPF-2.75 is much fewer than others in Figure 7 (full cell). NVPF 2.75 and 3 are the highlights of the paper, so I believe the number of compared cycles should be same with others.

We fully agree with the comment, and this was corrected in the revised version.

e) In line 175, page 8, it was stated that: “Interestingly, the $\Delta x = 2.75$ and $\Delta x = 3$ samples stand as an exception, since they never convert back into the pristine Amam phase as shown in Fig. 4b.”. However, in Fig 4b, the XRD of $\Delta x = 3$ sample was not shown.

We truly apologize for continuously bothering the referee by not providing both the electrochemical and structural data for the $\Delta x=3$ sample. The revised version contains all the requests formulated by the referee.

f) The authors applied soft XAS V-L edge and O-K edge to explain the V charge compensation mechanism. It is well known that soft-XAS strongly depends on surface properties and the results are strongly controlled by surface behavior. As far as I am concerned, hXAS should be a more accurate description of charge compensation mechanism.

The referee is right about the surface sensitivity of the soft XAS when measured in photoelectron yield. In this case, we are using TFY data which contains spectral information as deep as 100 nm (approximately) and therefore is probing the bulk behavior. To confirm our conclusions are not influenced by an experimental artifacts, the pre-edge of the O-K edge spectra can be compared to the pre edge feature of the V K-edge data since the same states are probed, i.e. O(2p) - V(3d) hybridized states. The figure below shows the evolution of the V K-edge spectra during the removal of the 2 first Na from NVPF (blue is the pristine, green Na2VPF and the red NaVPF) highlighting the pre edge features. Three peaks are observed in the pristine, 5466 eV, 5468 eV and 5469.5 eV. During oxidation, new peaks appear at 5469.5 and 5471 eV together with the progressive disappearance of the low energy peaks. This evolution is exactly similar to what is observed in the O-K edge spectra. The only difference is that the new features are appearing at lower energy in the O K edge data. This is due to the valence shift of the V as exemplified below. Overall, our data seems to correlates perfectly with hXAS data hence proving the reliability of our sXAS measurements.

Redacted

Having addressed the well spotted scientific questions and suggestions raised by the referees, and keeping their positive overall reaction in mind, we hope that you will now accept the revised version for publication in Nature Communications.

Reviewers' comments:

Reviewer #1 (Remarks to the Author):

After answering to all the queries from these reviewers in a satisfactory way, I think that the work is worth publishing in its revised form. The work is interesting for the battery community and represents an advance in the knowledge of fluorophosphate phases.

Reviewer #2 (Remarks to the Author):

I suggest to reject its publication in Nature Communications.

Thank the authors for addressing my comments. However, my concern about the electrochemical performance section in the manuscript has not been properly addressed. In the section of studying NVPF materials, the authors claimed that NVPF-3 presents the best performance by using 3rd Na. However, in the revision of NVPF/C full cells, the authors demonstrated that NVPF-2.5/C presents the highest energy density of 451 Wh kg⁻¹ when cycled between 4.3-2.0 V, NVPF-2.75/C shows the highest energy density of 458 Wh kg⁻¹ between 4.5-0 V (See Fig. 8), but in Fig. S12, NVPF-2.6/C delivered the highest energy density. Apparently, the authors' description about the optimal battery performance is very confusing. Which one is the best choice among NVPF-2.5, NVPF-2.6 and NVPF-2.75? Further, none of the three optimal materials is NVPF-3. In Fig. 2, NVPF-3 presents the highest reversible capacities and excellent cyclic stability, but NVPF-3/C full cells show a lower capacity and the lowest capacity retention. Please give some explanation. Last, numerous efforts have been devoted to improving the performance of NVPF cathodes, the authors are suggested to compare the electrochemical performance of current NVPF-3 with its peers to evaluate the standing among literatures.

Reviewer #3 (Remarks to the Author):

In the revised manuscript, the author clarified the electrolyte problem, provided necessary XRD data, provided the V K edge XANES in comparison with soft XAS data.

However, it seems that the newly provided XRD data has inconsistency issues. The NVDF 2.75 and 3.0 share a similar structure (the higher symmetry I4/mmm). So XRD of NVDF 2.75 and 3.0 are quite similar (Figure 4a), which is consistent and expected. XRD of NVDF 2.75 and 3.0 discharged to 3V are similar too (Figure S8), which is consistent and expected. However, the XRD of NVDF 2.75 and 3.0 discharged to 1 V (newly provided) are very different (Figure 4b, new shoulder peaks, significant peak splitting and very different peak intensity ratios). Thus, it seems that the NVDF 2.75 and 3.0 discharged to 1V have different structures, which is basically inconsistent to the author's claim that they are all I4/mmm. Refinement results and table for NVDF 3.0 discharged to 1V is necessary to justify the XRD results.

Reviewer #2

QUESTION-1: Thank the authors for addressing my comments. However, my concern about the electrochemical performance section in the manuscript has not been properly addressed. In the section of studying NVPF materials, the authors claimed that NVPF-3 presents the best performance by using 3rd Na. However, in the revision of NVPF/C full cells, the authors demonstrated that NVPF-2.5/C presents the highest energy density of 451 Wh kg⁻¹ when cycled between 4.3-2.0 V, NVPF-2.75/C shows the highest energy density of 458 Wh kg⁻¹ between 4.5-0 V (See Fig. 8), but in Fig. S12, NVPF-2.6/C delivered the highest energy density. Apparently, the authors' description about the optimal battery performance is very confusing. Which one is the best choice among NVPF-2.5, NVPF-2.6 and NVPF-2.75? Further, none of the three optimal materials is NVPF-3. In Fig. 2, NVPF-3 presents the highest reversible capacities and excellent cyclic stability, but NVPF-3/C full cells show a lower capacity and the lowest capacity retention.

RESPONSE: We think that there is some miss-reading of the data and we want to raise the referees' attention that our electrochemical data of the NVPF materials are first reported in half-cell and then in full Na-ion cells with different rankings performance-wise. So for pedagogical reasons we are addressing below point by points the various statements within this question.

1) "Our claim of NVPF-3 presents the best performance by using the 3rd Na in NVPF structure".

This is true as conveyed in Figure 2 for Na-half cells: Indeed, when we discharge the NVPF-2.0, NVPF-2.25, NVPF-2.5, NVPF-2.75, NVPF-3.0 samples down to 1.0 V, the NVPF-3.0 does show the highest capacity, while showing equal cycling performance as the others. More importantly, we had also emphasized (page 4, line 97-98) that "However, this ~40 % gain in capacity narrows down to solely a ~15% benefit in energy density because most of the extra capacity occurs at low potential (Fig. S4)". Now turning to full cells our NVPF-3 does not rank any longer first as mentioned in the paper and explained next.

2) "The authors demonstrated that NVPF-2.5/C presents the highest energy density of 451 Wh kg⁻¹ when cycled between 4.3-2.0 V, NVPF-2.75/C shows the highest energy density of 458 Wh kg⁻¹ between 4.5-0 V (See Fig. 8), but in Fig. S12, NVPF-2.6/C delivered the highest energy density. Apparently, the authors' description about

the optimal battery performance is very confusing. Which one is the best choice among NVPF-2.5, NVPF-2.6 and NVPF-2.75?"

We are now dealing with full cells (Fig. 8 and Fig. S12) whose performances is determined by their cycling conditions. Here, we have used two variable namely limiting voltage and cell balancing (ratio of positive to negative electrode $R=m+/m-$), that will of course affects the cell energy density. In figure 8, the cells are assembled with an $R = 2$ using a cycling voltage of 4.3-2V (Fig. 8a), the NVPF-2.5 shows the highest energy density. And using a cycling voltage of 4.3-0 V, so under such conditions (which make use of the low voltage plateau for extra safety), the NVPF-2.75 sample is indeed the best. Now moving to figure S12, the referee must note that the cycling voltage is not any longer 4.3- 0 V but 4.3 – 2V, with here a final optimization of the cell by adjusting the R value. In that case, the best energy density by fixing the cycling voltage between 4.3 and 2V can be achieved for NVPF-2.6 with an optimized R of 1.98, for which the amount of the extra Na removed is perfectly adjust to match the amount of Na irreversibly consumed at the negative (SEI) during the first charging of Na-full cells.

So, in short we personally don't see any inconsistency in our data. The difference in reported numbers are simply the results of different cell assemblies (R) or cycling conditions. Playing with either R or cell cycling voltages to adjust the energy density or cell safety risks is quite common when optimizing a system.

Lastly, we would like to mention that if we replace carbon by another negative electrode showing larger irreversible capacities than carbon (alloys and others) in that case NVPF-2.8, NVPF-2.9 or even NVPF-3 could be the composition for achieving the highest energy density. In short, assembling full cells with negative electrodes of increasing irreversible capacity will require the removal of greater amount of Na to obtain the best energy density performance, that is a positive electrode having a greater Na reservoir, hence the preference for compositions approaching NVPF-3. This stresses again that there is not a single or universal composition for achieving the highest energy density. That is the reason why the amount of removed Na can fluctuate depending upon the chemistry, the adjustment of the mass balance for either assembling cells with either safety or energy advantages and so on.

Whatever, in spite of the veracity of our results, we can empathize with the referee that the huge amount of electrochemical data tested in various conditions for reaching the highest energy density, which is achieved by utilizing our fundamental findings of activating the 3rd Na in NVPF can be view as confusing. Therefore, to reduce some burdens to the reader we have modified the text at a few places to simplify the reading as well as add some warning notes in figure captions.

QUESTION-2: Last, numerous efforts have been devoted to improving the performance of NVPF cathodes, the authors are suggested to compare the electrochemical performance of current NVPF-3 with its peers to evaluate the standing among literatures.

RESPONSE: We found this statement somewhat awkward and don't understand why the reviewer claims that our peers have achieved better performances without giving any hints or papers that we could have missed. So we decided to go into a thorough survey of today's literature, considering both Na-based NVPF or Na-related phases.

Results of our survey regarding polyanionic compounds are presented in the table below for which we give the compound formula, the energy density of these compounds deduced by multiplying the voltage by capacity on the basis of Na half-cell data, and the accompanying references. It can be note that performance-wise our NVPF-3 sample shows the highest energy density (563 Wh/kg). The one that is coming the closest is $\text{Na}_3\text{V}_2(\text{PO}_4)_2\text{O}_2\text{F}$ (540 Wh/kg, colored yellow in the table). However, a serious drawback for this

Sample	Chemical formula	Energy density (Wh kg ⁻¹)	Reference
1	$\text{Na}_3\text{V}_2\text{O}_2(\text{PO}_4)_2\text{F}$	388	[1]
2	$\text{Na}_3\text{V}_2\text{O}_{2x}(\text{PO}_4)_2\text{F}_{3-2x}$	323	[2]
3	$\text{Na}_3(\text{VOPO}_4)_2\text{F}$	447	[3]
4	$\text{Na}_3(\text{VPO}_4)_2\text{F}_3$	461	[3]
5	$\text{Na}_3(\text{VOPO}_4)_2\text{F}$	421	[4]
6	$\text{Na}_3\text{V}_2(\text{PO}_4)_2\text{O}_2\text{F}$	486	[5]
7	$\text{Na}_3\text{V}_2(\text{PO}_4)_2\text{FO}_2$	540	[6]
8	$\text{Na}_3\text{V}_2(\text{PO}_4)_2\text{F}_{2.5}\text{O}_{0.5}$	434	[7]
9	$\text{Na}_4\text{MnV}(\text{PO}_4)_3$	400	[8]
10	$\text{Na}_3\text{V}_2(\text{PO}_4)_2\text{O}_{1.6}\text{F}_{1.4}$	518	[9]
11	$\text{Na}_3\text{V}_2(\text{PO}_4)_2\text{F}_3$	563	Our work

compound as compared to NVPF-3 is that it cannot serve as Na reservoir since V is at the 4+ state in the pristine phase as opposed to 3+ in our compound. This explains why only two Na⁺ can be extracted from $\text{Na}_3\text{V}_2(\text{PO}_4)_2\text{O}_2\text{F}$ as compared to 3 from NVPF-3 in the first charge process. Hence, it cannot be used as Na-reservoir in full Na-ion cells to compensate for the irreversible capacity lost during the first cycle in the full cell. This means by implementing $\text{Na}_3\text{V}_2(\text{PO}_4)_2\text{O}_2\text{F}$ in full cells ($\text{Na}_3\text{V}_2(\text{PO}_4)_2\text{O}_2\text{F}/\text{C}$) we will get reduced energy density than the values reported for half cells which will show at least a 10% penalty in energy density performance than in NVPF-2.6/C full cells.

1. Xu M, Xiao P, Stauffer S, Song J, Henkelman G, Goodenough JB. Theoretical and Experimental Study of Vanadium-Based Fluorophosphate Cathodes for Rechargeable Batteries. *Chem. Mater.* 26, 3089-3097 (2014).

- Kumar PR, Jung YH, Lim CH, Kim DK. $\text{Na}_3\text{V}_2\text{O}_{2x}(\text{PO}_4)_2\text{F}_{3-2x}$: a stable and high-voltage cathode material for aqueous sodium-ion batteries with high energy density. *J. Mater. Chem. A* 3, 6271-6275 (2015).
- Zhao JM, Mu LQ, Qi YR, Hu YS, Liu HZ, Dai S. A phase-transfer assisted solvo-thermal strategy for low-temperature synthesis of $\text{Na}_3(\text{VO}_{1-x}\text{PO}_4)_2\text{F}_{1+2x}$ cathodes for sodium-ion batteries. *Chem. Commun.* 51, 7160-7163 (2015).
- Qi YR, Mu LQ, Zhao JM, Hu YS, Liu HZ, Dai S. Superior Na-Storage Performance of Low-Temperature-Synthesized $\text{Na}_3(\text{VO}_{1-x}\text{PO}_4)_2\text{F}_{1+2x}$ ($0 \leq x \leq 1$) Nanoparticles for Na-Ion Batteries. *Angewandte Chemie-International Edition* 54, 9911-9916 (2015).
- Guo J-Z, *et al.* High-Energy/Power and Low-Temperature Cathode for Sodium-Ion Batteries: In Situ XRD Study and Superior Full-Cell Performance. *Adv. Mater.* 29, (2017).
- Bianchini M, Xiao P, Wang Y, Ceder G. Additional Sodium Insertion into Polyanionic Cathodes for Higher-Energy Na-Ion Batteries. *Adv. Energy Mater.* 7, (2017).
- Broux T, *et al.* Temperature Dependence of Structural and Transport Properties for $\text{Na}_3\text{V}_2(\text{PO}_4)_2\text{F}_3$ and $\text{Na}_3\text{V}_2(\text{PO}_4)_2\text{F}_{2.5}\text{O}_{0.5}$. *Chem. Mater.* 30, 358-365 (2018).
- Chen F, *et al.* A NASICON-Type Positive Electrode for Na Batteries with High Energy Density: $\text{Na}_4\text{MnV}(\text{PO}_4)_3$. *Small Methods* 0, 1800218 (2018).
- Li C, *et al.* High-energy nanostructured $\text{Na}_3\text{V}_2(\text{PO}_4)_2\text{O}_{1.6}\text{F}_{1.4}$ cathodes for sodium-ion batteries and a new insight into their redox chemistry. *J. Mater. Chem. A* 6, 8340-8348 (2018).

As a result of this survey, we could not spot a single recent paper that reports performances of Na-Half cells based either on NVPF or derived polyanionic compounds that is achieved by utilizing our fundamental findings of activating the 3rd Na in NVPF. Even more importantly, our trick of the third Na, make NVPF-X the best material today for Na-ion full cells. This is the reason why our patented discovery has been licensed by the TIAMAT Company that develops the NVPF-X/C Na-ion technology.

Reviewer #3

QUESTION: However, it seems that the newly provided XRD data has inconsistency issues. The NVPF 2.75 and 3.0 share a similar structure (the higher symmetry $I4/mmm$). So XRD of NVPF 2.75 and 3.0 are quite similar (Figure 4a), which is consistent and expected. XRD of NVPF 2.75 and 3.0 discharged to 3V are similar too (Figure S8), which is consistent and expected. However, the XRD of NVPF 2.75 and 3.0 discharged to 1 V (newly provided) are very different (Figure 4b, new shoulder peaks, significant peak splitting and very different peak intensity ratios). Thus, it seems that the NVPF 2.75 and 3.0 discharged to 1V have different structures, which is basically inconsistent to the author's claim that they are all $I4/mmm$. Refinement results and table for NVPF 3.0 discharged to 1V is necessary to justify the XRD results.

RESPONSE: We thank the referee for his/her attentive examination of data. The XRD pattern that is plotted for NVPF3-discharged to 1 V on top to figure 4b (orange pattern) indeed presents a different peak intensity ratio than the other patterns, and also presents shoulders that could indicate that the symmetry is no longer tetragonal. First, let us clarify the questions of symmetry. For compound NVPF-2.0, NVPF-2.25 and NVPF-2.5 all discharged to 1 V, we have used the *Amam* orthorhombic space group with lattice parameters $a \approx 9.04 \text{ \AA}$, $b \approx 9.05 \text{ \AA}$ and $c \approx 10.77 \text{ \AA}$. This cell perfectly indexes also the pattern of a NVPF that was directly discharged to 1 V (never charged), as can be seen on Figure 1a below. When the same pattern is refined using a tetragonal cell with $I4/mmm$ space group and lattice parameters $a \approx 6.40 \text{ \AA}$, $b \approx 6.40 \text{ \AA}$ and $c \approx 10.77 \text{ \AA}$ (Figure 1b), some reflections are not taken into account, see for instance the one marked with yellow arrows on Figure 1b. Therefore, the presence or absence of these peaks can serve as an indicator for the symmetry.

Figure 1: Refinement of NVPF discharged to 1 V (never charged) in *Amam* (a) and $I4/mmm$ (b) space groups.

Coming back at the XRD patterns plotted in Figure 4b in the paper, we now reexamine all patterns according to these characteristic peaks. The patterns are shown in Figure 2 here, and the dotted vertical lines indicate the positions of peaks characteristics of the Amam cell. From this, it can be seen that neither NVPF2.75-dis 1V nor NVPF3-dis 1V present those peaks, therefore their patterns can be indexed in the tetragonal $I4/mmm$ unit cell (both cells are related by $\sqrt{2} \times \sqrt{2} \times 1$).

Figure 2a: Part of the XRD patterns of the NVPF-2 to NVPF-3 samples discharged to 1 V.

Figure 2b: 11BM synchrotron XRD pattern of the two samples NVPF-3 sent to 11BM (red: original data; blue: new data obtained during the review process in October 2018); the XRD patterns are in full agreement except the $2\theta = 7.4^\circ$ shoulder that comes from sample evolution.

Figure 2a also explains the shoulders spotted by the referee. One can observe that these are at the exact same position (around $2\theta = 7.4$ deg.) as the peaks from NVPF-2.75 discharged to 1V; the sample therefore either slightly evolved while being sent and measured on synchrotron or the cell was not at the equilibrium. This is further confirmed since we sent another sample to the synchrotron when we received referee's 3 comments. It appears that our new set of data is fully consistent with the original one, with the exception of this shoulder at 7.4° which is even more pronounced (Figure 2b). Indeed, 11BM Argonne beamline was shut down in September 2018 and the sample waited for three weeks before being measured. This feature is therefore clearly not related to the same phase. We are however confident that the structural analysis carried out on the main phase is not affected by this feature.

Next, as requested by the referee, we present in Figure 3 below (and in Figure S7 of the paper) the refinement of the "NVPF3-discharged to 1 V" pattern, and the table of resulting structural parameters in Table 1 below (and Table S6 in the main paper).

Figure 3: Rietveld refinement and resulting structure of NVPF3-dis 1 V.

Table 1. Structural parameters of NVPF-3 discharged to 1 V determined from the Rietveld refinement of the synchrotron 11BM XRD data.

$\text{Na}_x\text{V}_2(\text{PO}_4)_2\text{F}_3$ ($x = 3.36(1)$ from refinement), $I4/mmm$ space group,
 $a = 6.5080(3) \text{ \AA}$, $c = 10.7233(7) \text{ \AA}$,
 $V = 454.17(4) \text{ \AA}^3$, $VIZ = 227.08 \text{ \AA}^3$, $R_{\text{Bragg}} = 6.18 \%$, $\chi^2 = 1.03$

atom	Wyckoff site	x	y	z	B (\AA^2)	Occ
V1	4d	0	0	0.1897(3)	1.42(5)	1
P1	4e	0	0.5	0.25	3.61(15)	1
O1	16n	0	0.3194(7)	0.1644(4)	3.14(14)	1
F2	4e	0	0	0.3736(9)	6.2(2)	1
F1	2a	0	0	0	6.2(2)	1
Na1	8h	0.2698(5)	0.2698(5)	0	2.9(2)	0.83(1)
Na2	16l	0.416	0.231	0	2.9(2)	0.01(2)

It appears clearly that the pattern can be fit with the tetragonal $I4/mmm$ space group. Moreover, the above refinement sheds also light on the referee's question about the intensity ratio. The relative intensity ratio we observe is reliable since the new NVPF-3 dis 1V sample we sent to synchrotron presents a fully reproducible pattern (see Figure 2b above). This change in intensities comes from the Na distribution among the different Na sites. Indeed, we can notice that the NVPF-3 dis 1V has almost all sodium atoms distributed on the Na1 site (8h Wyckoff site). At the contrary, for NVPF-2.75 dis 1V, Na is distributed mainly on Na1 but also some Na is on the 16l Na2 site. To further confirm this explanation, we present in Figure 4 three simulations. The structural models used for simulating the three patterns differ only by the distribution of the 3.2 sodium per f.u. (6.4 per unit cell) among the 8h (Na1) and 16l (Na2) sites. One can clearly see that the intensity ratio between peaks at 7.4° and 8.5° is reversed when some Na is transferred from Na2 to Na1, and vice-versa.

Figure 4 XRD simulations of structural models of NVPF-2.75 dis 1 V that only differ by the distributions of 3.2 Na (6.4 per unit cell) between two crystallographic sites Na1 and Na 2.

To further confirm this explanation, we present in Figure 4 three simulations. The structural models used for simulating the three patterns differ only by the distribution of the 3.2 sodium per f.u. (6.4 per unit cell) among the 8h (Na1) and 16l (Na2) sites. One can clearly see that the intensity ratio between peaks at 7.4° and 8.5° is reversed when some Na is transferred from Na2 to Na1, and vice-versa.

We have done our best to satisfactorily answer the referee remaining questions and hope that you will now accept the revised version of the paper for publication.

REVIEWERS' COMMENTS:

Reviewer #3 (Remarks to the Author):

In the revised manuscript, the authors proposed that the differences on Na site occupancy (Na1 8h or Na2 16i) lead to the significant differences of XRD profiles for NVPF-3_1V and NVPF-2.75_1V. Simulated XRD based on this rationale could basically demonstrate the experimental trend of peak splitting and ratio behavior for NVPF-3_1V and NVPF-2.75_1V (Figure S7 fourth and fifth panel). Personally, I would recommend acceptance of this manuscript, since the structural analysis is quite complete now. It should be noted that the error for NVPF-3_1V refinement is significantly larger than refinements of other panels, especially for higher degree peaks. Additionally, the author's claim regarding the reason for the additional 7.4 degree shoulder peak for is NVPF-3_1V also likely.

Reviewer's comments:

Reviewer #3 (Remarks to the Author):

In the revised manuscript, the authors proposed that the differences on Na site occupancy (Na1 8h or Na2 16i) lead to the significant differences of XRD profiles for NVPF-3_1V and NVPF-2.75_1V. Simulated XRD based on this rationale could basically demonstrate the experimental trend of peak splitting and ratio behavior for NVPF-3_1V and NVPF-2.75_1V (Figure S7 fourth and fifth panel). Personally, I would recommend acceptance of this manuscript, since the structural analysis is quite complete now. It should be noted that the error for NVPF-3_1V refinement is significantly larger than refinements of other panels, especially for higher degree peaks. Additionally, the author's claim regarding the reason for the additional 7.4 degree shoulder peak for is NVPF-3_1V also likely.

Response: We appreciate the positive comments from the reviewer.

Thanks the notes from reviewer. We noticed that the error between observed and calculated pattern is relative larger than the other samples. However, this error does not affect our claim that the pattern of NVPF-3.0 discharged to 1V sample can be indexed in the tetragonal I4/mmm unit cell, which the reviewer agreed. Regarding the reason for the additional 7.4 degree shoulder peak, indeed, we suppose that the NVPF-3.0 discharged to 1 V sample either slightly evolved while being sent and measured on synchrotron or the cell was not at the equilibrium. Combined TEM, EPR and operando XAS studies are needed to further exploring the sodium-driven local structural changes as we indicated in the discussion part.